# The nucleolus is the site for inflammatory RNA decay during infection

Taeyun A. Lee[1], Heonjong Han[1,2,3], Ahsan Polash ●[4], Seok Keun Cho[1], Ji Won Lee[5], Eun A. Ra[1], Eunhye Lee[1], Areum Park[1], Sujin Kang[1], Junhee L. Choi[1], Ji Hyun Kim[6], Ji Eun Lee[6,7], Kyung-Won Min[5,8], Seong Wook Yang[1], Markus Hafner ●[4], Insuk Lee[2], Je-Hyun Yoon ●[8] ✉, Sungwook Lee ●[3] ✉ & Boyoun Park ●[1] ✉

Inflammatory cytokines are key signaling molecules that can promote an immune response, thus their RNA turnover must be tightly controlled during infection. Most studies investigate the RNA decay pathways in the cytosol or nucleoplasm but never focused on the nucleolus. Although this organelle has well-studied roles in ribosome biogenesis and cellular stress sensing, the mechanism of RNA decay within the nucleolus is not completely understood. Here, we report that the nucleolus is an essential site of inflammatory pre-mRNA instability during infection. RNA-sequencing analysis reveals that not only do inflammatory genes have higher intronic read densities compared with non-inflammatory genes, but their pre-mRNAs are highly enriched in nucleoli during infection. Notably, nucleolin (NCL) acts as a guide factor for recruiting cytosine or uracil (C/U)-rich sequence-containing inflammatory pre-mRNAs and the Rrp6-exosome complex to the nucleolus through a physical interaction, thereby enabling targeted RNA delivery to Rrp6-exosomes and subsequent degradation. Consequently, *Ncl* depletion causes aberrant hyperinflammation, resulting in a severe lethality in response to LPS. Importantly, the dynamics of NCL post-translational modifications determine its functional activity in phases of LPS. This process represents a nucleolus-dependent pathway for maintaining inflammatory gene expression integrity and immunological homeostasis during infection.

Inflammatory cytokines are immune regulatory proteins that promote innate and adaptive immune responses to various pathogens[1,2]. Due to this role, high expression levels of inflammatory genes are associated with various autoimmune diseases and tumor growth[1,3–7], and thus

their RNA expression must be carefully controlled. Regnase-1 and Roquin have been known to be an essential protein involved in targeted inflammatory mRNA degradation in the cytoplasm[8–10]; however, the mechanisms underlying intact pre-mRNA decay in the nucleus

[1]Department of Systems Biology, College of Life Science and Biotechnology, Yonsei University, Seoul, South Korea. [2]Department of Biotechnology, College of Life Science and Biotechnology, Yonsei University, Seoul, South Korea. [3]Division of Tumor Immunology, Research Institute, National Cancer Center, Goyang, South Korea. [4]Laboratory of Muscle Stem Cells and Gene Regulation, National Institute for Arthritis and Musculoskeletal and Skin Disease, National Institutes of Health, Bethesda, MD, USA. [5]Department of Biology, College of Natural Sciences, Gangneung-Wonju National University, Gangneung, South Korea. [6]Department of Health Sciences and Technology, Samsung Advanced Institute for Health Sciences and Technology, Sungkyunkwan University, Seoul, South Korea. [7]Samsung Genome Institute (SGI), Samsung Medical Center, Seoul, South Korea. [8]Department of Biochemistry and Molecular Biology, Medical University of South Carolina, Charleston, SC, USA. ✉e-mail: yoonje@musc.edu; swlee1905@ncc.re.kr; bypark@yonsei.ac.kr

remain to be elucidated. Although several findings suggest the mammalian RNA surveillance pathway rapidly degrades pre-mRNA in the nucleus[11–13], most studies have observed the pre-mRNA turnover pathway in the entire nucleus but never focused on specific subcellular or subnuclear organelles or documented the process under infectious conditions.

The nucleolus, a well-known subnuclear body, is the traditional site for ribosomal RNA (rRNA) synthesis and ribosomal subunit assembly, controlling the critical downstream functions of protein synthesis, which can affect fundamental biological processes such as growth or survival[14–16]. Data also demonstrate the nucleolus has roles beyond ribosome biogenesis, including transmitting signals in response to genome instability and nutrient alteration, regulating lifespan, and responding to viral infection[17–21]. Recent studies suggest extracellular adhesive cues impact nucleolar structure and subsequent rRNA transcription, which is dependent on the cytoskeleton[22]. In particular, changes in nucleolar morphologies occur in *Caenorhabditis elegans* against bacterial infection[23], implying that nucleolar structure and function may be linked to host immune response during infections.

Here, we report a role for the nucleolus as the essential subnuclear site of inflammatory pre-mRNA decay. In particular, inflammatory genes have higher intronic read densities compared with non-inflammatory genes and their pre-mRNAs are highly enriched within nucleoli during infection. Importantly, nucleolin (NCL) acts as a guide factor for recruiting cytosine or uracil (C/U)-rich sequence-containing pre-mRNAs and the Rrp6-exosome complex to the nucleolus through a physical interaction, thereby enabling targeted RNA delivery to Rrp6-exosomes and subsequent degradation. Consequently, *Ncl*-depleted mice show augmented inflammatory RNA levels, resulting in a severe lethality in response to bacterial lipopolysaccharide (LPS). Our study reveals why inflammatory pre-mRNAs accumulate in the nucleolus and how NCL- and Rrp6-dependent decay regulates inflammatory gene expression under infectious conditions in vitro and in vivo. These findings reinforce the novel concept that the nucleolus is an essential site of inflammatory pre-mRNA turnover and elucidated how immune homeostasis can be maintained by controlling inflammatory gene expression.

## Results

### Nucleoli are fused in later stages of infection

The morphological dynamics and plasticity that subcellular organelles exhibit in response to environmental stimuli reflect their underlying functional optimization. Thus, we initially examined whether infection affects subcellular organelle morphology in RAW 264.7 macrophages. While evaluating morphological changes of subcellular organelles in response to LPS-triggered Toll-like receptor 4 (TLR4) signaling, we intriguingly observed nucleoli, which are typically irregular in shape and multinucleolate under normal conditions, exhibited a fused morphology, increased size, and significantly decreased number at later time points (Supplementary Fig. 1a). Moreover, time-lapse monitoring of green fluorescent protein (GFP)-tagged fibrillarin (FBL), a nucleolus-specific marker for dense fibrillar components (DFCs), showed similar structural alterations of the nucleolus (Fig. 1a, b, and Supplementary Movie 1). This was confirmed by other nucleolar markers for fibrillar centers (FCs) and granular components (GCs) (Supplementary Fig. 1b). Following recovery from LPS treatment, the nucleoli reverted back to their typical shape and size with a multinucleolate phenotype similar to that in unstimulated RAW 264.7 cells (Fig. 1c and Supplementary Fig. 1c). These phenomena were also detected in LPS-stimulated bone marrow-derived macrophages (BMDMs) and dendritic cells (BMDCs), as well as in RAW 264.7 cells treated with Poly(I:C) or CpG-DNA, a ligand of TLR3 or TLR9 (Fig. 1d, e). Consistent with these results, infection by *Escherichia coli* (*E. coli*), a

Gram-negative bacterium, also induced nucleolar fusion at the later stages of infection (Fig. 1f). These results indicate that infection is linked to changes in nucleolar morphology.

### Inflammatory pre-mRNAs are enriched in nucleoli during infection

To understand the relationship between changes in nucleolar structure and infection, we focused on nucleolar RNAs because nucleolus-enriched RNA polymerase II (Pol II) transcripts have been reported to affect nucleolar structure and eventually regulate rRNAs synthesis[24]. We first examined the expression level of rRNAs within the nucleolus during LPS stimulation. However, no considerable changes in 18S, 28S, or 47S rRNA levels were observed at any of the LPS treatment times examined (Supplementary Fig. 2a). We further investigated whether protein-coding transcripts are present in the nucleolus or whether their subcellular abundance and distribution are influenced by a time course of LPS stimulation. To test this, we purified cytoplasmic (CA), nucleoplasmic (NP), and nucleolar (NL) fractions from RAW 264.7 cells treated with LPS for 0, 2, 6, 12, or 15 h and then verified the purity of all fractions by immunoblotting with antibodies against specific protein markers (Supplementary Fig. 2b). To enable efficient protein-coding transcript detection, highly abundant rRNAs were removed from total RNA. We then analyzed the RNA-seq dataset by using the Integrative Genomics Viewer (IGV), which allowed us to determine the read density of exons and introns in each fraction. IGV displayed considerable reads of introns as well as exons of ubiquitously expressed protein-coding genes (e.g., *B2m*, *Actb*, *Ldha*, and *Pgk1*) in both the nucleoplasmic and nucleolar fractions, all of which were similar patterns of read density across time courses of LPS stimulation (Supplementary Fig. 2c). In addition, we selectively examined the read density of inflammatory genes (*Il6*, *Il1a*, or *Ccl2*) in each fraction. Intriguingly, we observed that the intact *Il6*, *Il1a*, or *Ccl2* pre-mRNAs, not intron remnants after splicing, were substantially enriched in the nucleolus after LPS stimulation, with high levels occurring at 2 or 6 h and a gradual decline after 12 h (Fig. 2a).

To accurately assess global changes in distributions or expression levels of non-inflammatory or inflammatory pre-mRNAs across different samples of each fraction, the RNA-seq data was subject to quantile normalization after conducting exon-intron split analysis. We further applied more stringent selection criteria to determine a significant pattern of genes induced by LPS and independently sorted protein-coding genes ($n = 5097$) into constitutive and non-inducible ($0 <$ fold-change [FC] $\leq 1$, $n = 3636$), moderately inducible ($1 < FC \leq 2$, $n = 1023$), and highly inducible ($FC > 2$, $n = 438$) gene categories. Based on gene ontology biological process, we classified 115 genes belonging to the category of highly inducible genes as inflammatory genes, prompting us to examine the spatiotemporal dynamics of intron and exon reads between non-inflammatory (constitutive and non-inducible, $n = 3636$) and inflammatory genes ($n = 115$) (Supplementary Data 1). Notably, in the nucleolus, high amounts of inflammatory pre-mRNAs were observed after 2 h of LPS stimulation, whereas non-inflammatory pre-mRNAs were relatively low and unchanged throughout the LPS time course (Fig. 2b). Moreover, inflammatory pre-mRNAs were more enriched in the nucleolus compared to the nucleoplasm; however, this difference was not observed in non-inflammatory pre-mRNAs (Fig. 2c). This nucleolar enrichment of inflammatory pre-mRNAs peaked at 2 and 6 h of LPS exposure and gradually decreased, which was similar to the phased pattern of nucleoplasmic pre-mRNAs and cytoplasmic mRNAs (Fig. 2b). Of note, a higher percentage of intron reads were found in inflammatory RNA libraries from the nucleoplasmic fractions throughout LPS stimulation relative to non-inflammatory RNA libraries, demonstrating the proportion of intron to exon reads in inflammatory genes was much higher than in non-inflammatory genes (Fig. 2d). These findings indicate that the intronic region of inflammatory genes may be involved in its nucleolar transport and function.

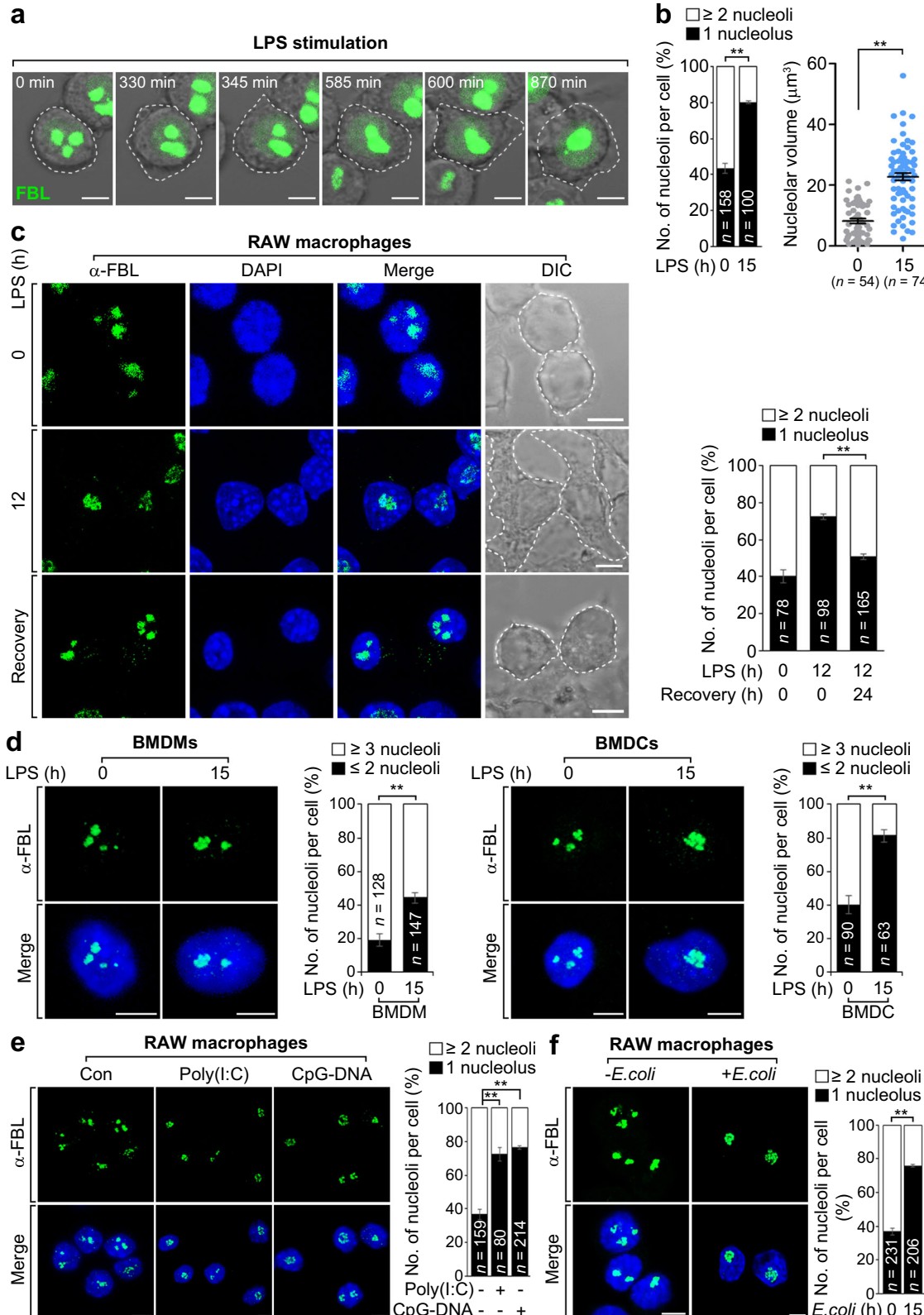

## Nucleolar localization of inflammatory pre-mRNAs is linked to their gene expression

To verify the nucleolar enrichment of inflammatory pre-mRNAs, we performed RNA-fluorescence in situ hybridization (RNA-FISH) analysis on cells stimulated with LPS for 12 h using RNA probes that specifically recognize the intronic region of inflammatory RNAs. We found that the inflammatory *Il1b*, *Il6*, *Ccl2*, and *Cxcl2* pre-mRNA foci were

preferentially positioned close to nucleoli or overlapped with the rim of the nucleolus, whereas non-inflammatory *Hmga1* pre-mRNA foci were positioned away from the nucleolus (Fig. 3a, white arrows). To confirm this, we measured the distance between the centers of nucleolus and the pre-mRNA foci of *Hmga1*, *Il1b*, *Il6*, *Ccl2*, and *Cxcl2*. We observed that the inflammatory pre-mRNA foci were much closer to nucleoli than non-inflammatory pre-mRNA foci (Fig. 3b). To further

**Fig. 1 | Nucleoli are fused in later stages of infection. a** Still images from time-lapse sequences showing nucleolar fusion in FBL-GFP-expressing RAW 264.7 macrophages by analyzing patterns of FBL through LPS (80 ng ml⁻¹) stimulation. Dashed lines delineate the borders of same cell for chasing nucleolar fusion in a cell. **b** Graphs showing the percentage of cells with one or more than two nucleoli (left) or the volume of nucleoli (right). Dot plot is presented as means ± s.e.m. **c** Images showing the nucleolar morphology in RAW 264.7 cells after 12 h LPS stimulation or recovery from LPS treatment. The dashed line delineates the borders of cells. Graphs showing the percentage of cells with one or more than two nucleoli. For

details on experimental design of recovery from LPS stimulation, see Supplementary Fig. 1c or Methods. **d–f** Images showing nucleolar fusion at the indicated times of LPS stimulation in BMDMs or BMDCs (**d**), and in RAW 264.7 cells stimulated by CpG-DNA (1 μM) or Poly(I:C) (100 μg ml⁻¹) for 18 h (**e**), or infected with *E. coli* for 15 h (**f**). Graphs representing the percentage of cells with indicated nucleoli numbers per cell. *n*, total cells counted. Green FBL, blue DAPI. Scale bars, 5 μm. *P* values are determined by unpaired two-tailed *t* test. \*\**P* < 0.01. All data are representative of three independent experiments and are presented as means ± s.d. in (**b–f**). Source data are provided as a Source Data file.

exclude any potential artifacts from passive streaming due to robust inflammatory RNA production in response to LPS, we generated RAW 264.7 cells stably expressing a 5′-*Myc*-tagged *Il1b* or *Cox6a2* minigene composed of intron and exon regions corresponding to an inflammatory or non-inflammatory gene, respectively (Fig. 3c, top). Quantitative reverse transcription-PCR (RT-qPCR) using primer pairs that specifically recognize the 5′-*Myc* tag or intron region was performed to assess RNA levels in nucleolar fractions and thus exclude endogenous inflammatory RNAs induced by LPS. Indeed, the RNA level of *Myc*-tagged *Il1b*, but not that of *Myc*-tagged *Cox6a2*, was increased in the nucleolar fraction of LPS-exposed RAW 264.7 cells (Fig. 3c, bottom).

Previous reports suggest specific sites located closer to the nucleolus tend to be transcriptionally inactive[25–27]. Thus, we examined the spatial preferences of inflammatory pre-mRNA foci with RNA polymerase II (RNA-pol II) during LPS stimulation. To test this, we combined the FISH signals for inflammatory or non-inflammatory pre-mRNAs with immunofluorescence for nucleoli (FBL-GFP) and active RNA-pol II (anti-pS2; Ser-2 phosphorylation of RNA-pol II is a marker of active RNA-pol II). Intriguingly, inflammatory *Il1b* and *Il6* pre-mRNA foci overlapped with nucleoli but were positioned away from active RNA-pol II regions, while non-inflammatory *Hmga1* pre-mRNA foci were preferentially positioned in active RNA-pol II regions and excluded from the nucleolar compartment (white arrows in Fig. 3d, i and iii). We also observed inflammatory pre-mRNA foci-pS2 contacts in the nucleoplasm, but none of which were positioned in the nucleolus (yellow arrows in Fig. 3d, ii and iv). In addition, TDP-43, a protein marker for the *Il6* RNA splicing activating compartment (InSAC)[28], was found to be separated from the nucleolus (Supplementary Fig. 3a, white arrows). Consistent with the confocal microscopy results, TDP-43 was detectable in the nucleoplasmic fraction, but not in the nucleolar fraction (Supplementary Fig. 3b). These results indicate that the nucleolus-enriched inflammatory pre-mRNAs are unlikely to be involved in transcription or splicing.

We further investigated whether the destruction of nucleolar structure affects inflammatory RNA expression. Notably, nucleolar disruption induced by depletion of nucleophosmin 1 (*Npm1*) or treatment with etoposide, which is often used to cause nucleolar destruction or distortion[29,30], resulted in a significant increase in LPS-induced inflammatory RNA levels (*Il1b*, *Il6*, *Cxcl2*, *Ccl2*, *Ccl4*, and *Ccl5*) (Fig. 3e). This was not due to any effect on transcription as no differences were observed in the promoter activity of *Il6* in *Npm1*-depleted or etoposide-treated RAW 264.7 cells (Supplementary Fig. 3c). Moreover, inflammatory *Il6* pre-mRNA foci were away from active RNA-pol II (Supplementary Fig. 3d, white arrows) and their size and fluorescence intensity were significantly increased in *Npm1*-depleted cells compared to control cells (Supplementary Fig. 3e). This indicates the nucleolus is involved in regulating inflammatory pre-mRNA stability rather than transcription. To support these results, we separated nucleoplasmic and nucleolar fractions from 12 h LPS-stimulated RAW 264.7 cells and then examined the half-life of inflammatory pre-mRNAs by using primers for intra-intronic regions after treatment with the transcription inhibitor, actinomycin D (Act.D). Importantly, the turnover rate of *Il1b*, *Il6*, *Cxcl2*, *Ccl2*, *Ccl4*, and *Ccl5* pre-mRNAs was much faster in the nucleolar fraction than in the nucleoplasmic fraction (Fig. 3f). This was also confirmed

by using primers for intron-exon regions of *Il1b* and *Il6* (Supplementary Fig. 3f). Taken together, our findings suggest that inflammatory pre-mRNAs are transported to the nucleolus, which is involved in their RNA destabilization and gene expression.

## NCL is essential for inflammatory pre-mRNA decay

To next explore how the inflammatory pre-mRNAs are located to the nucleolus during LPS stimulation, we screened for potential RNA-binding proteins that enable the transport of these RNAs to the nucleolus using nucleolar proteome datasets[31–33]. Three independent studies revealed that pumilio RNA-binding family member 3 (PUM3), partner of NOB1 (PNO1), nucleolar protein 6 (NOL6), and nucleolin (NCL) are RNA-binding proteins that not only reside primarily in nucleoli but are also able to translocate to different subcellular locations in response to alterations in environmental condition or nucleolar structure[34–37] (Supplementary Fig. 4a). This prompted us to investigate the effect of loss of *Pum3*, *Pno1*, *Nol6*, or *Ncl* expression on inflammatory RNA expression levels. Interestingly, inflammatory pre-mRNA expression level of *Il1b*, *Il6*, *Cxcl2*, *Ccl2*, and *Ccl5*, but not *Ccl4*, elicited by 12 h LPS stimulation was significantly increased only in *Ncl*-depleted RAW 264.7 cells (Supplementary Fig. 4b). Moreover, their mRNA levels were highly increased in *Ncl*-depleted cells, an effect that was reversible by overexpression of GFP-tagged NCL (NCL-GFP) (Fig. 4a). As a result, the secretion level of *Il1b*, *Il6*, *Ccl5*, but not *Ccl4*, was increased in *Ncl*-depleted cells (Supplementary Fig. 4c). These effects were not due to transcriptional activities because of similar *Il6* promoter activity in *Ncl*-depleted or NCL-GFP-overexpressing RAW 264.7 cells (Supplementary Fig. 4d). In addition, although most studies have showed the role of NCL for rRNA processing in epithelial, endothelial cells, or fibroblasts but not in immune cells[38–40], we found no significant difference in the level of rRNA species between control and *Ncl*-depleted RAW 264.7 cells following 12 h LPS stimulation (Supplementary Fig. 4e). The RT-PCR results with the indicated primer pairs demonstrate that *Ncl* depletion induced both pre-mRNA and mRNA level of *Il1b*, *Il6*, *Cxcl2*, and *Ccl2*, while northern blot analysis showed a similar increase in both types of *Il1b* and *Il6* RNA (Fig. 4b and Supplementary Fig. 5a). Consistent with these in vitro results, mice depleted of *Ncl* in liver- or lung-resident cells exhibited a significant increase in *Il1b*, *Il6*, or *Ccl5* pre-mRNA and mRNA levels in liver or lung tissues (Fig. 4c and Supplementary Fig. 5b), which in turn led to higher total serum levels of IL-6 and CCL5 and susceptibility to LPS-induced lethality (Fig. 4d and Supplementary Fig. 5c). Similar to the previous results, no differences in *Ccl4* expression were observed (Fig. 4c and Supplementary Fig. 5c). These data suggest that NCL is linked to the instability of inflammatory pre-mRNAs rather than to attenuation of RNA splicing. To support these results, we examined the half-life of inflammatory pre-mRNAs after treatment with Act.D, in *Ncl*-depleted RAW 264.7 cells after 12 h LPS stimulation. The turnover rate of *Il1b*, *Il6*, *Cxcl2*, *Ccl2*, and *Ccl5*, but not *Ccl4*, was much slower in *Ncl*-depleted cells, whereas the corresponding pre-mRNAs destabilized rapidly in control cells (Fig. 4e). Thus, our findings raise the possibility that NCL may modulate the stability of *Il1b*, *Il6*, *Cxcl2*, *Ccl2*, and *Ccl5* during late-stage LPS stimulation by binding RNA substrates that possess specific core sequences located within their intronic regions.

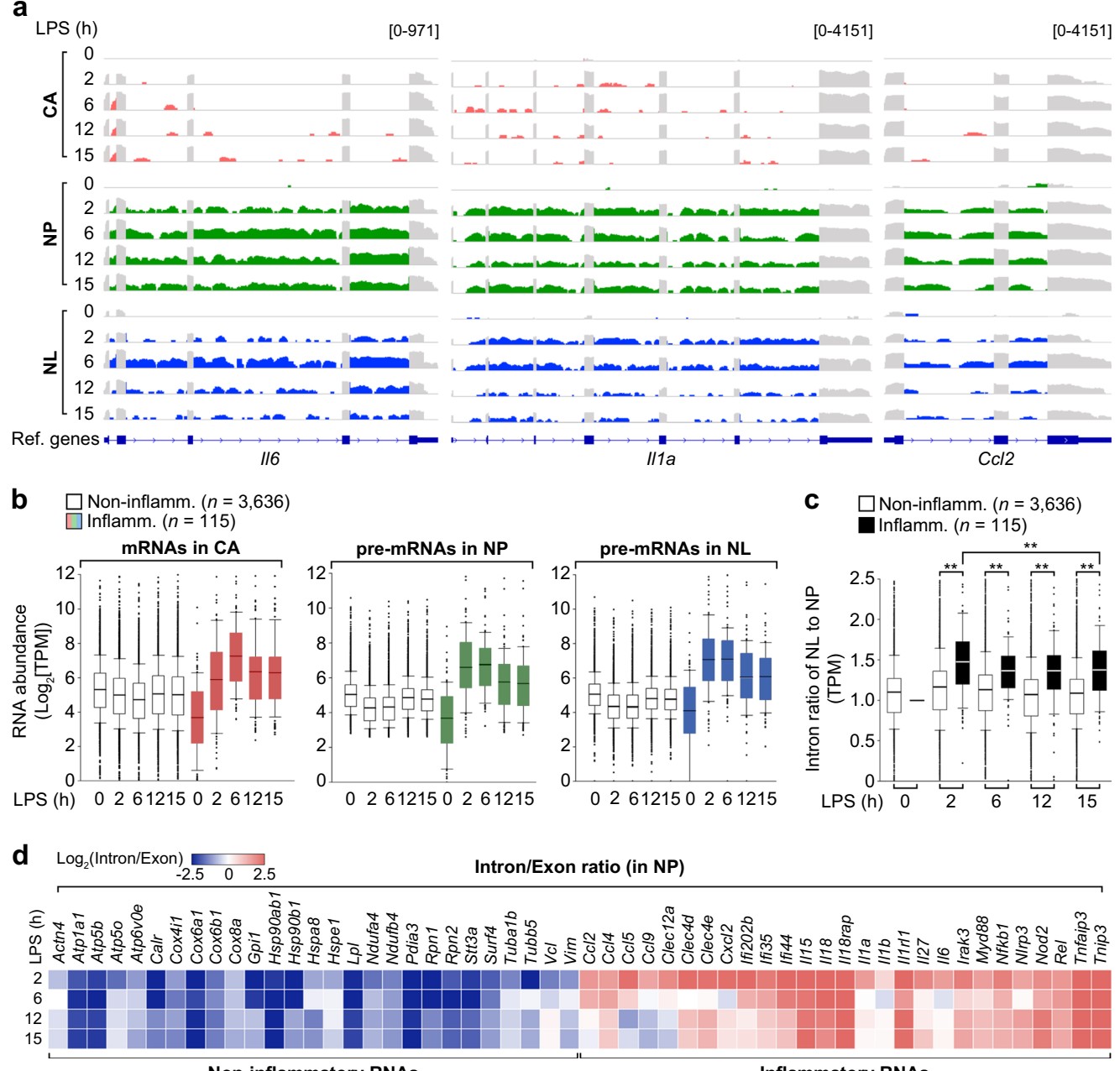

**Fig. 2 | Nucleoli contain inflammatory pre-mRNAs during infection. a** IGV displaying the read density of inflammatory genes (*Il6*, *Il1a*, and *Ccl2*) in CA, NP, and NL fractions at 0, 2, 6, 12, or 15 h of LPS stimulation. Read coverage tracks in red, green, and blue indicate inflammatory intronic reads in the CA, NP, and NL, respectively. Grays denote inflammatory exonic read coverage in all fractions. **b** Comparison of changes in mRNA and pre-mRNA levels of non-inflammatory (Non-inflamm., *n* = 3636) and inflammatory (Inflamm., *n* = 115) genes in CA, NP, and NL fractions at 0, 2, 6, 12, or 15 h of LPS stimulation. For RNA-seq analysis, StringTie, exon-intron split analysis, and quantile normalization was applied to each fraction. For details on RNA-seq, see Methods. **c** Time-course changes in the NL/NP ratio (nucleolar

enrichment) of intronic reads between non-inflammatory and inflammatory genes. For (**b**, **c**), box plots represented 5th, 25th, 50th, 75th and 95th percentiles, with median values labeled. **\*\****P* < 0.01 (Two-tailed Mann–Whitney *U* test). **d** Heat map representing the ratio of individual intronic to exonic reads between non-inflammatory (*n* = 27) and inflammatory (*n* = 27) genes in NP fractions from RAW 264.7 macrophages stimulated with LPS at the indicated times. Red or blue colors represents high or low proportion of intron to exon reads, respectively. All data are representative of three independent experiments. Source data are provided as a Source Data file.

## NCL binding through a C/U-rich sequence determines nucleolar targeting and instability of inflammatory pre-mRNAs

To thus identify the consensus RNA-binding motif(s) (RBM) of NCL capable of determining its functional specificities in an unbiased manner, we subjected LPS-treated RAW 264.7 cells to photoactivatable ribonucleoside-enhanced crosslinking and immunoprecipitation (PAR-CLIP) using an anti-NCL antibody followed by genome-aligned reads

after deep-sequencing of the recovered RNAs with PARalyzer[41,42] (Supplementary Fig. 6a). This analysis identified 10,548 NCL-binding sites, of which 46% were mapped to protein-coding RNAs and 82.6% originated from intron sequences (Fig. 5a and Supplementary Data 2). Intriguingly, our motif analysis revealed that a pyrimidine (cytosine or uracil; C/U)-rich consensus sequence consisting of over 20 nucleotides is present within the intronic regions of various inflammatory pre-

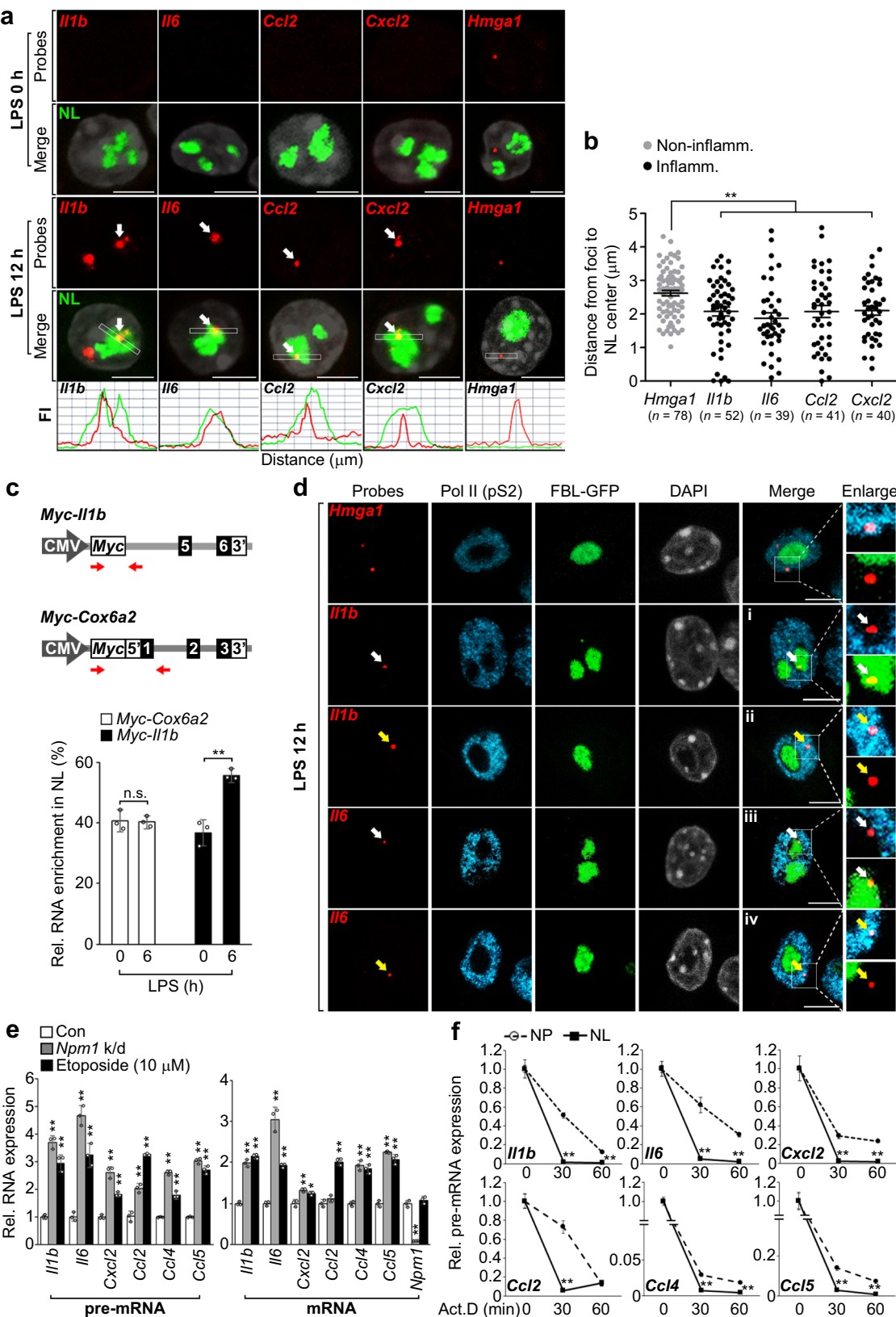

mRNAs (42 of 115 inflammatory genes), but not in *Ccl4* (Fig. 5b and Supplementary Data 1). We also observed NCL binding to the *Il1b*, *Il6*, *Cxcl2*, *Ccl2*, and *Ccl5* pre-mRNAs, but not to the *Ccl4* or non-inflammatory *Cox6a2* or *Gapdh* pre-mRNAs (Supplementary Fig. 6b). These patterns were reminiscent of prior results from *Ncl* depletion studies showing the expression and stability of *Il1b*, *Il6*, *Cxcl2*, *Ccl2*, and

*Ccl5*, but not *Ccl4* pre-mRNA (Fig. 4a, c, e). Indeed, analyzing the RNA-seq dataset with LPS-inducible genes ($n = 438$), which were > 2 fold higher after LPS stimulation, and sorting 106 NCL-targeted and 332 non-targeted genes based on the presence or absence of C/U-rich sequences in their introns demonstrated that C/U-rich inflammatory pre-mRNAs were much more highly enriched in the nucleolus at 2 h

**Fig. 3 | Nucleolar localization of inflammatory pre-mRNAs is involved in their gene expression. a** RNA-FISH showing nucleolar enrichment of inflammatory pre-mRNAs in 12 h LPS-stimulated RAW 264.7 cells. Graphs represent quantification of colocalization of 47S rRNA (green; unprocessed rRNA as a specific nucleolar RNA marker) with indicated inflammatory or non-inflammatory pre-mRNAs (red). Arrows indicate nucleolar enrichment of inflammatory pre-mRNAs. DAPI, gray. Scale bars, 5 μm. **b** Graphs showing the distance between the centers of nucleolus and the pre-mRNA foci of *Hmga1*, *Il1b*, *Il6*, *Ccl2*, and *Cxcl2*. *n*, total foci counted. Data are presented as means ± s.e.m. **c** Top, schematic representation showing the 5′-*Myc*-tagged *Il1b* and *Cox6a2* minigene constructs. Closed or open boxes represent exons or untranslated regions in 3′ or 5′, respectively, and connecting lines indicate introns. Red arrows depicting primer pairs recognizing either *Myc* or intron region. CMV CMV promoter, *Myc Myc* sequence. Bottom, graph plotting RNA levels transcribed from minigenes in nucleolar fractions of LPS-stimulated RAW 264.7 cells.

**d** RNA-FISH showing the spatial preferences of inflammatory pre-mRNAs between the nucleolus and active transcription sites in FBL-GFP-expressing RAW 264.7 cells under 12 h LPS stimulation. White or yellow arrows indicate inflammatory pre-mRNAs located in the nucleolus or active RNA-pol II regions (Pol II [pS2]), respectively. White boxes delineate the area of enlargement showing localization of pre-mRNA foci. DAPI, gray. Scale bars, 5 μm. **e** qPCR showing inflammatory RNA levels in *Npm1*-depleted or etoposide-treated RAW 264.7 cells during 12 h LPS stimulation. **f** Graphs showing the half-life of inflammatory pre-mRNAs in nucleoplasmic or nucleolar fractions from 12 h LPS-stimulated RAW 264.7 cells after Act.D treatment for indicated time points. This is measured by qPCR using intra-intron primer pairs. *P* values are determined by unpaired two-tailed *t* test. *$P < 0.05$; **$P < 0.01$. n.s. not significant. All data are representative of three independent experiments and are presented as means ± s.d. in (**c**, **e**, **f**). Source data are provided as a Source Data file.

post-LPS stimulation (1.25 compared to 1.1), but eventually decreased after 6 h (Supplementary Fig. 6c and Supplementary Data 1). We also confirmed the binding ability of NCL with this C/U-rich sequence by performing RNA electrophoretic mobility shift assay (EMSA) using *Il6* or *Il1b* RNA probes with a 48-nucleotide oligo possessing 20 or 21 repeats of a C/U-rich sequence positioned in introns 2 or 6 corresponding to *Il6*-[C/U]$_{20}$ or *Il1b*-[C/U]$_{21}$, respectively (Fig. 5c). NCL bound to *Il6*-[C/U]$_{20}$ or *Il1b*-[C/U]$_{21}$ with high affinity and the $K_d$ value of both RNAs was similar to that of the NCL recognition element (NRE), a known strong binding sequence of NCL[43] (Fig. 5d, e). An alternative method using a fluorescence anisotropy-based RNA-binding assay[44], which is useful for evaluating protein-RNA-binding affinity, also showed that NCL-bound *Il1b* and *Il6* RNAs with high affinity, as characterized by a low $K_d$ value (Supplementary Fig. 6d). Importantly, mutational analysis of core consensus sequences demonstrated that reducing the number of C/U motifs within the 48-nt *Il1b* RNA segments by substitution of four or eight C/Us with adenine (A), which is one of the purines (*Il1b*-[C/U]$_{17}$ and *Il1b*-[C/U]$_{13}$), resulted in a significant reduction in NCL-binding affinity (Fig. 5c–e). We also found that the $K_d$ value of *Il1b*-[C/U]$_{13}$ was ~40-fold higher on average compared to wild-type *Il1b*-[C/U]$_{21}$, suggesting that NCL binding depends on the number of C or U.

Given that NCL-bound inflammatory pre-mRNAs possessing a C/U-rich RBM within intronic regions exhibit nucleolar enrichment in response to LPS, we next investigated whether these pre-mRNAs are able to bind NCL and subsequently transported to the nucleolus. To test this, we generated a deletion mutant of the 5′-*Myc*-tagged *Il1b* minigene which lacks a C/U-rich RBM positioned within intron 6 (*Myc-Il1b* ΔC/U-rich) and examined its ability to bind NCL and RNA levels in nuclear fractions (Fig. 5f, top). Unlike the wild-type *Myc-Il1b* minigene, *Myc-Il1b* ΔC/U-rich exhibited reduced NCL binding and RNA level in nucleolar fractions, supporting the hypothesis that NCL binding of inflammatory pre-mRNAs is essential for nucleolar targeting (Fig. 5f, bottom).

## NCL delivers inflammatory pre-mRNAs to the Rrp6-exosome for their degradation

Our findings demonstrate that NCL is capable of binding inflammatory pre-mRNAs through a C/U-rich RBM in introns, thus enabling them to translocate to the nucleolus and possibly facilitate instability. This led us to consider the contribution of NCL in directly regulating the instability of nucleolar inflammatory pre-mRNAs. As NCL itself does not possess RNase activity, we tried to identify the exonuclease that is involved in degradation of NCL-bound inflammatory pre-mRNAs. To do this, we conducted a large-scale purification with RAW 264.7 cells stably expressing GFP-tagged full-length NCL (FL-NCL-GFP) using a retroviral system to achieve maximal pull-down efficacy. However, ectopic expression of full-length NCL results in relatively lower expression levels of NCL possibly due to disruption of viral assembly through the acidic region[45,46]. We thus used a deletion mutant of NCL

that lacked acidic domains (NCLΔAS-GFP) for identifying the exonuclease, since this deletion mutant considerably affected nucleolar enrichment and instability of *Il1b* and *Il6* pre-mRNAs, although to a lesser extent than full-length NCL (Supplementary Fig. 7a–d). Mass spectrometric analysis showed that a significant number of polypeptides, including the nucleus-specific exosome component Rrp6 and RNA helicases (DDX5 and DHX36, DEAD-Box Helicase 5 and DEAH-Box Helicase 36) were associated with NCL in response to LPS exposure (Fig. 6a). Particularly, since Rrp6 is capable of binding the core exosomes in the nucleus, thus facilitating RNA degradation[13,47,48], we explored the roles of the Rrp6 in decay of NCL-bound inflammatory pre-mRNAs. Consistent with the results from large-scale purification, endogenous NCL associated with Rrp6 as well as the core exosome components, Rrp43 and Rrp46, and these interactions were detectable only in LPS-stimulated RAW 264.7 cells (Fig. 6b and Supplementary Fig. 8a). Moreover, depletion of each exosome component resulted in a significant increase in both RNA levels of *Il1b* and *Il6* that was reversible by Rrp6 overexpression (Fig. 6c and Supplementary Fig. 8b). In a line with previous reports showing that Rrp6-exosomes require RNA-binding proteins for RNA degradation because Rrp6 itself is not capable of binding RNAs stably[49–53], we also found that the level of *Ccl4* RNA, which did not exhibit NCL binding or NCL-mediated instability, was affected by *Rrp6* overexpression or depletion (Supplementary Fig. 8b, c, compared with Fig. 4a, c, e). This raised the hypothesis that NCL may be a guide factor essential for delivering NCL-bound inflammatory RNA substrates to the Rrp6-exosomes for degradation in the nucleolus. Although Rrp6-exosomes have been known to be mostly localized throughout the nucleoplasm under normal physiological conditions[47,54], we found that Rrp6, Rrp43, and Rrp46 predominantly translocated into the nucleolus from the nucleoplasm and colocalized with NCL following LPS stimulation, which was specifically observed in RAW 264.7 macrophages and BMDMs, but not in HeLa or NIH3T3 cells (Fig. 6d and Supplementary Fig. 8d, e). Of note, *Ncl*-depleted RAW 264.7 cells showed a failure in Rrp6 binding capability of *Il1b* and *Il6* pre-mRNAs as well as its translocation into the nucleolus even after LPS treatment (Fig. 6e, f). Taken together, our findings suggest that NCL is critical for not only targeting Rrp6-exosomes and inflammatory pre-mRNAs to the nucleolus via a physical interaction at later phases of LPS stimulation but also for guiding those pre-mRNAs to the Rrp6-exosome complex for degradation.

## Dynamics of NCL PTMs determines its functional activity during infection

Based on our data showing NCL association with Rrp6 and core exosomes only occurs after LPS exposure, we hypothesized NCL may undergo post-translational modifications (PTMs) during infection. In the same context, NCL has been reported to be affected by PTMs in response to various stimuli[55–57]. In particular, NCL localizes exclusively to the nucleus when threonine 76 (Thr76) is dephosphorylated; however, NCL exhibits enhanced cytoplasmic localization when this site is

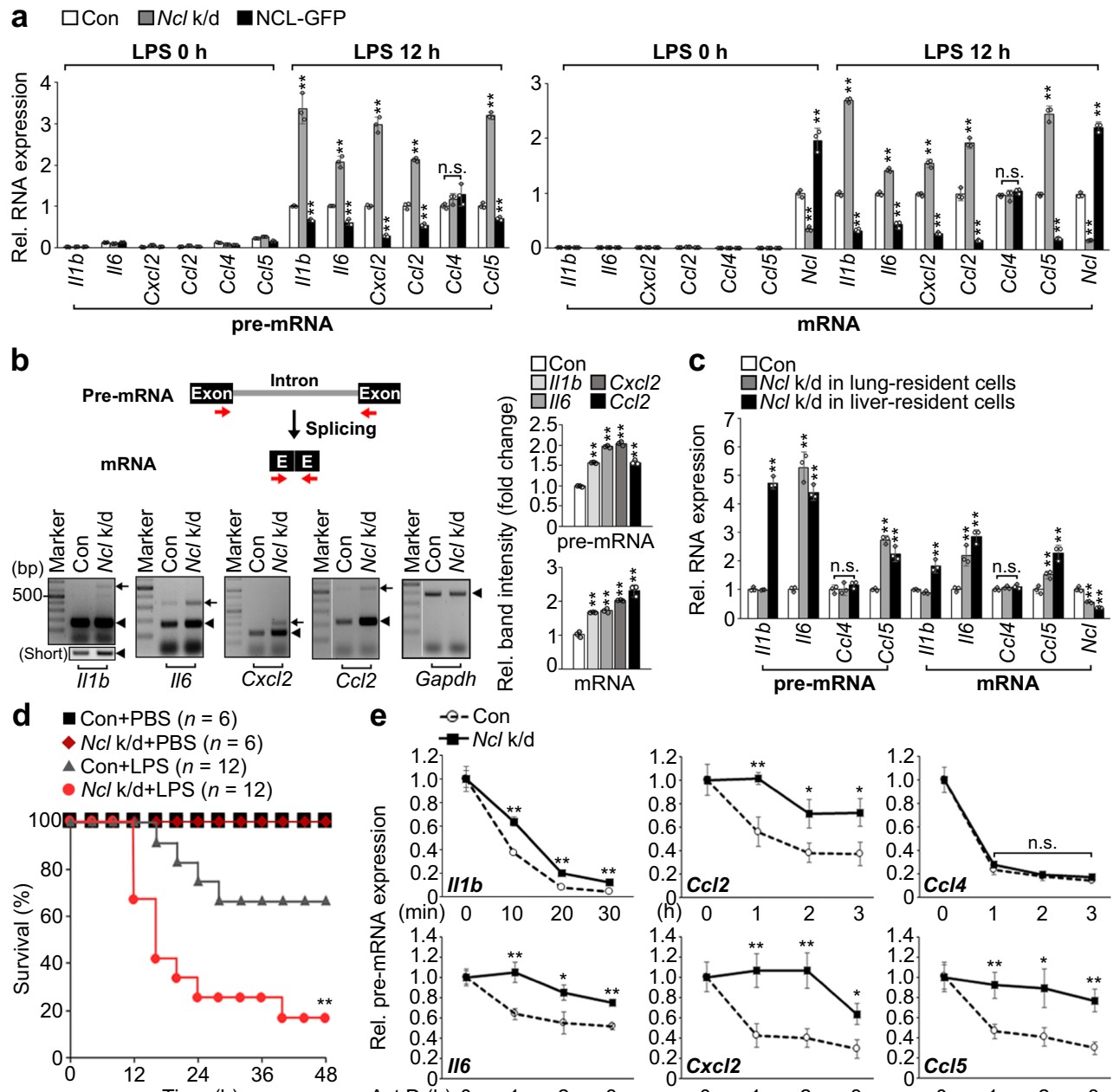

**Fig. 4 | NCL is a key factor essential for inflammatory RNA decay. a** Results of qPCR for inflammatory RNA levels in *Ncl*-depleted or overexpressed RAW 264.7 cells under normal condition or 12 h LPS stimulation. **b** Schematic representation showing the strategy of RT-PCR primer design to detect exon-intron junction (red arrows). Bottom gels, RT-PCR showing effects of *Ncl* depletion on pre-mRNA or mRNA levels of indicated inflammatory genes in 12 h LPS-stimulated RAW 264.7 cells. Black arrows or arrowheads indicate pre-mRNA or mRNA, respectively. Short indicates short-term exposure of the upper image. Right graphs representing quantification of RT-PCR results. **c** Analysis of inflammatory RNA levels in lung- or liver-resident cells from control or *Ncl*-depleted mice after 10 h of LPS challenge.

**d** Survival rate between control and lung-specific *Ncl*-depleted mice after LPS challenge. *n* represents the number of mice. **P < 0.001 (Two-tailed log-rank test). For details on experimental design of in vivo experiments, see Supplementary Fig. 5b or Methods. **e** Analyzing pre-mRNA half-life of indicated inflammatory genes at three time points after Act.D treatment in 12 h LPS-stimulated RAW 264.7 cells. *P* values are determined by unpaired two-tailed *t* test. *P < 0.05; **P < 0.01 (Student's *t* test). n.s. not significant. All data are representative of three independent experiments and are presented as means ± s.d. in (**a–c**, **e**). Source data are provided as a Source Data file.

phosphorylated (pNCL-Thr76)[58] (Fig. 7a). These observations led us to investigate whether the phosphorylation status of NCL changes based on the duration of LPS exposure. Although a relatively small amount of NCL was detectable in the cytoplasm of RAW 264.7 cells, the level of pNCL-Thr76 was significantly reduced in cytoplasmic fractions at 12 h of LPS treatment despite a slight increase in the level of total NCL (Fig. 7b, first, second panel, and second graph). Moreover, okadaic acid (OA), a protein phosphatase 2A (PP2A) inhibitor, restored pNCL-Thr76

localization in the cytoplasm, suggesting that NCL dephosphorylation occurred at 12 h of LPS stimulation (Fig. 7b, second panel and graph). Depletion of *Ppp2cb*, a catalytic subunit of PP2A, confirmed this result as a significant increase in the level of phosphorylated NCL was observed in LPS-stimulated RAW 264.7 cells lacking *Ppp2cb* (Supplementary Fig. 9a). Given that NCL is essential for translocating Rrp6-exosomes to the nucleolus through a physical interaction during late-stage LPS stimulation, we next examined the effect of LPS-induced NCL

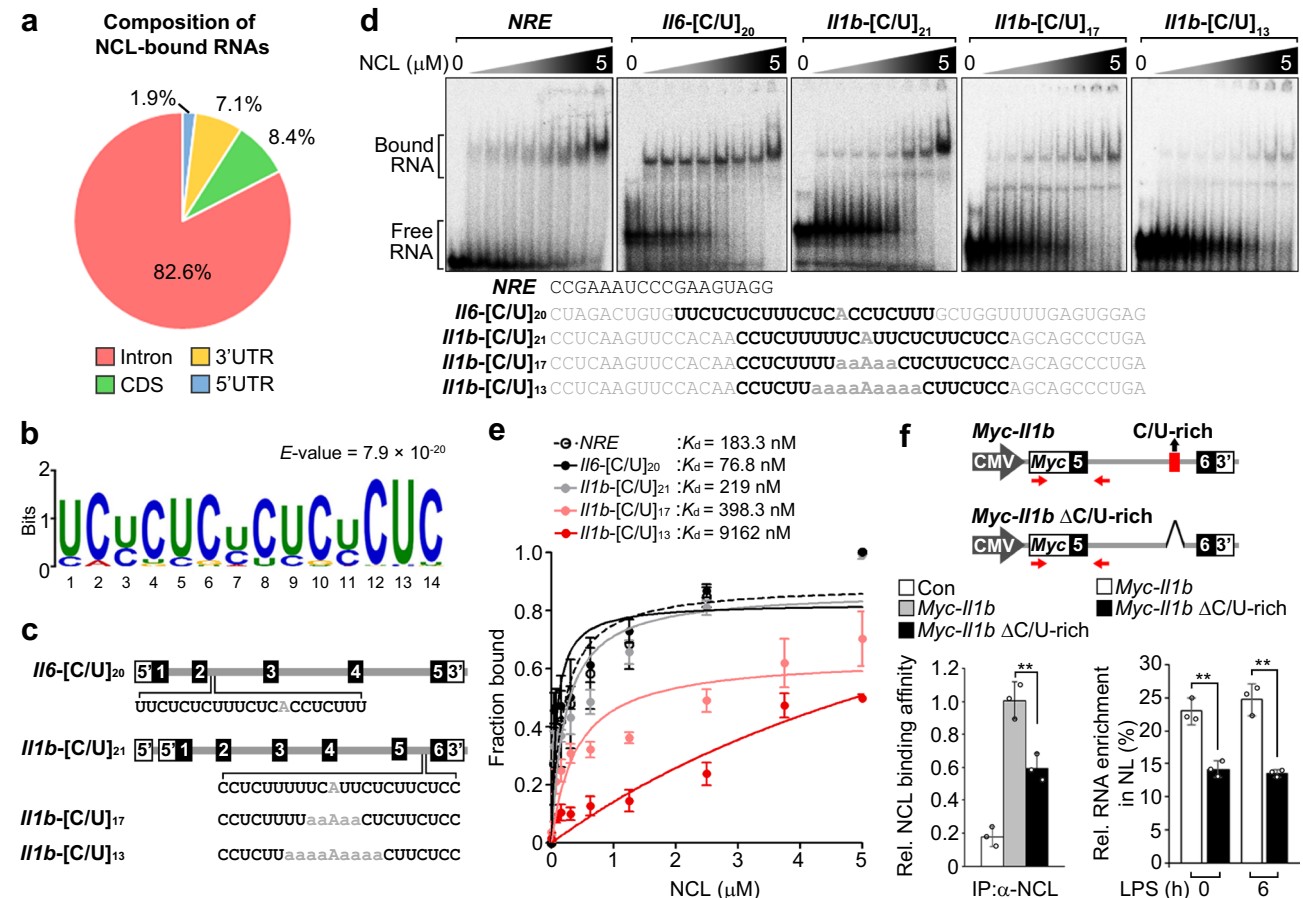

**Fig. 5 | NCL binding ability of inflammatory RNAs through a C/U-rich sequence determines its nucleolar targeting and instability. a** Pie chart presenting percentages of nucleolar RNA reads capable of binding NCL mapped to the indicated features based on PAR-CLIP analysis with nucleolar fractions of 12 h LPS-stimulated RAW 264.7 cells. For details on PAR-CLIP, see Methods. **b** Identification of a NCL-binding consensus motif analyzed by PAR-CLIP. **c** Schematic representation of wild-type or mutant *Il1b* or *Il6* RNA probes with reduced number of C or U for RNA EMSA. Numbers outside parentheses indicate the number of C or U positioning in the C/U-rich motif. Closed or open boxes represent exons or UTR in 3′ or 5′, respectively, and connecting lines indicate introns. **d**, **e** Gels for RNA EMSA showing NCL-binding affinity to the indicated RNA probes (**d**) and graphs showing $K_d$ values for the NCL-binding affinity to *Il1b* and *Il6* RNA probes (**e**). NRE is used as a positive control for assessing a strong binding affinity of NCL. **f** Top, schematic representation of wild-type or mutant *Myc*-tagged *Il1b* minigene lacking the C/U-rich sequences (*Myc-Il1b* ΔC/U-rich). Red box indicates C/U-rich sequences. Bottom, NCL-binding capacity (left) or nucleolar enrichment (right) of wild-type or *Myc-Il1b* ΔC/U-rich minigene. The NCL-binding capacity of *Myc-Il1b* or -ΔC/U-rich minigene was performed by RNA-IP-qPCR (RIP-qPCR) experiment. *P* values are determined by unpaired two-tailed *t* test. \*\**P* < 0.01 (Student's *t* test). All data are representative of three independent experiments and are presented as means ± s.d. in (**e**, **f**). Source data are provided as a Source Data file.

phosphorylation status on Rrp6 localization and its interaction with NCL. The nucleolar transport of Rrp6 was observed in 12 h LPS-stimulated cells but neither Rrp6 nor Rrp46 exhibited LPS-mediated nucleolar translocation in the presence of OA but instead retained their original nucleoplasmic localization (Fig. 7b, third panel and graph, Supplementary Fig. 9b, c). Moreover, interactions of NCL with Rrp6 were barely detectable in OA-treated cells even after 12 h LPS stimulation (Fig. 7c, first panels). We also noted that PP2A bound to NCL at 12 h of LPS stimulation and these interactions were disrupted in OA-treated cells, even with prolonged LPS stimulation (Fig. 7c, second panels).

Since NCL is sumoylated at lysine 296 (Lys296) within the NLS region, which maintains its nuclear localization[57] (Fig. 7a), we further examined the involvement of its sumoylation in response to LPS stimulation. We found that NCL sumoylation only occurred at 12 h of LPS stimulation, but was clearly limited in OA-treated cells even in the presence of LPS (Fig. 7d). Unlike wild-type NCL-GFP, the mutation of Lys296 to arginine (NCL-K296R-GFP) failed to be sumoylated even in LPS stimulation (Supplementary Fig. 9d). This mutant was also observed in the cytoplasm but with decreasing intensity in the nuclear fractions, which was confirmed by confocal microscopy

(Supplementary Fig. 9e, f). Moreover, the interaction between Rrp6 and NCL-K296R-GFP was disrupted, even in the presence of LPS (Fig. 7e). As a result, *Il6*, *Ccl2*, and *Ccl5* pre-mRNA expression levels were decreased only by overexpression of wild-type NCL-GFP, but not NCL-K296R-GFP (Fig. 7f). We also observed that substitution of Thr76 to alanine (NCL-T76A-GFP), a constitutively dephosphorylated status, maintained its sumoylation status even in unstimulated cells (Fig. 7g), suggesting that LPS-induced NCL dephosphorylation is critical for its subsequent sumoylation. In line with this result, the level of NCL-bound inflammatory pre-mRNAs was increased in OA-treated cells, which exhibited a high level of phosphorylated NCL as well as NCL overexpression failed to reduce *Il6* expression levels in OA-treated cells even in the presence of LPS (Supplementary Fig. 9g, h). Taken together, our results demonstrate that NCL PTMs occurs differently during LPS stimulation, which is required for nucleolar targeting of Rrp6 through NCL interaction and NCL-binding target RNA expression (Fig. 7h).

## Discussion

Inflammatory cytokines play an important protective role by promoting both innate and adaptive immune responses to various

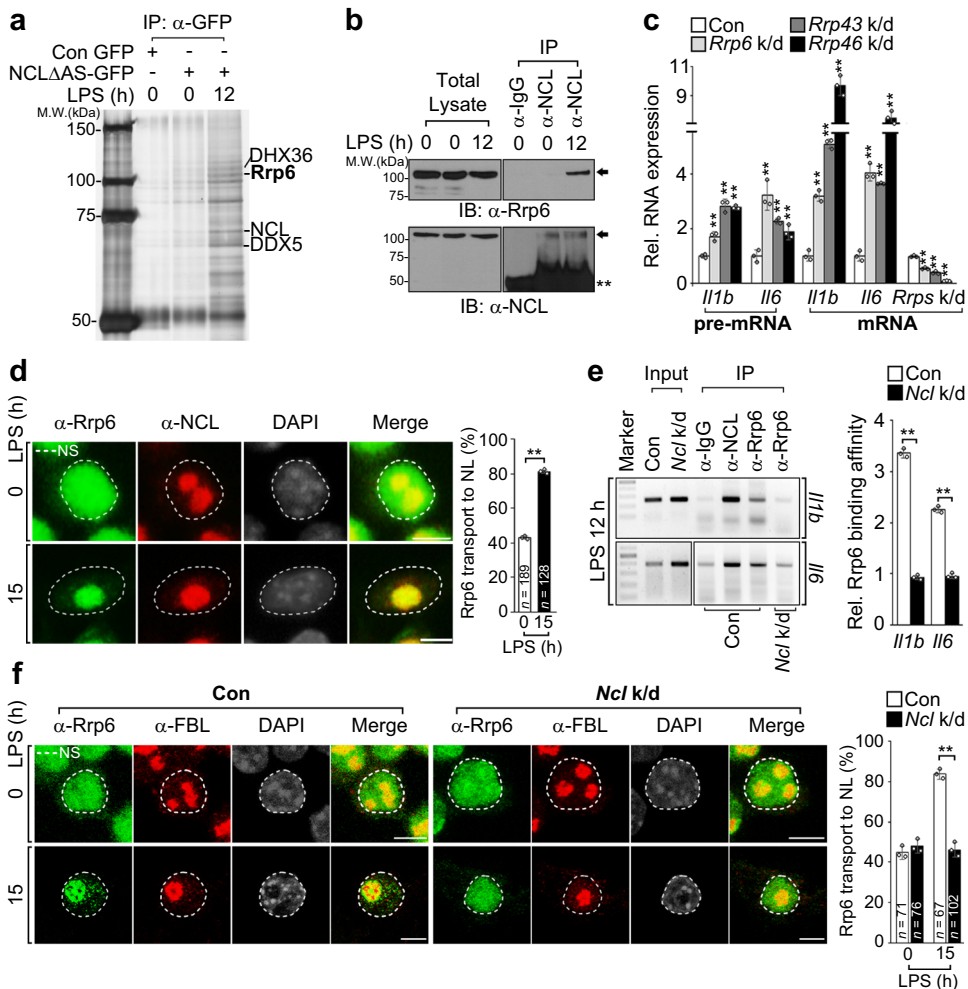

**Fig. 6 | NCL acts as a guide factor for delivering targeted inflammatory RNAs to the Rrp6-exosome and subsequent degradation in the nucleolus. a** Silver staining images showing NCL-binding proteins appeared only in LPS-stimulated NCLΔAS-GFP-expressing RAW 264.7 cells. **b** Interaction of NCL with Rrp6 occurred only by 12 h LPS stimulation. Arrows or asterisks represent the corresponding proteins immunoblotted by the indicated antibodies or IgG heavy chain, respectively. **c, d** Quantification of *Il1b* or *Il6* RNA levels in *Rrp6*, *Rrp43*, or *Rrp46*-depleted cells after 12 h LPS stimulation (**c**) or of nucleolar enrichment of Rrp6 in LPS-stimulated RAW 264.7 cells (**d**). The white dashed line delineates the border of nucleus (NS) based on DAPI. Graphs on right of the images representing percentage

of cells with Rrp6 nucleolar enrichment. *n*, total cells counted. Scale bars, 5 μm. **e, f** Effect of *Ncl* depletion on Rrp6 binding ability to *Il1b* or *Il6* pre-mRNA (**e**) and its nucleolar translocation (**f**) in 15 h LPS-stimulated RAW 264.7 cells. RNA-IP and images are quantified in right graph of each figure. Scale bars, 5 μm. *n*, total cells counted. For details of quantitative measurement of band intensity or colocalization, see Methods. *P* values are determined by unpaired two-tailed *t* test. \*\**P* < 0.01 (Student's *t* test). All data are representative of three independent experiments and are presented as means ± s.d. in (**c**–**f**). Source data are provided as a Source Data file.

infections; however, they can also be harmful if their expression and stability are dysregulated. Accordingly, failure of inflammatory cytokine RNA decay is closely associated with severe autoimmune diseases, including sepsis[8,59–61]. In addition, since inhibition of pre-mRNA degradation leads to increased mRNA production, it is important to discover the underlying mechanisms that regulate pre-mRNA metabolism and homeostasis. Here, we propose that the nucleolus is an essential organelle with functions beyond rRNA biogenesis, including playing a critical role in modulating the instability of inflammatory pre-mRNAs (Supplementary Fig. 10). Our results suggest that inflammatory pre-mRNAs are preferentially targeted and accumulate in the nucleolus at later times of infection and their expression is tightly controlled by NCL-Rrp6-exosome-dependent pre-mRNA decay occurred in the nucleolus. In particular, NCL is a critical guide factor that selectively binds inflammatory pre-mRNAs containing C/U-rich sequences within intronic regions, resulting in nucleolar targeting (Figs. 4, 5). Moreover, NCL is capable of associating with Rrp6-exosomes, subsequently facilitating their nucleolar targeting

to promote decay of targeted inflammatory pre-mRNAs in the nucleolus (Fig. 6).

Importantly, our study provides insight into how inflammatory pre-mRNA instability is mediated by the dynamics of NCL PTMs during infection (Fig. 7). NCL undergoes PP2A-mediated dephosphorylation followed by sumoylation and both modifications are essential for recruiting Rrp6-exosomes to NCL-bound inflammatory pre-mRNAs and tightly regulating their expression, possibly via nucleolar sequestration of robust inflammatory pre-mRNAs, to maintain immunological homeostasis at later stages of infection (Supplementary Fig. 10). Unlike our findings with the role of NCL in inflammatory pre-mRNA destabilization, several groups have shown NCL is involved in mRNA stability and binds to G-quadruplex or AU-rich sequences[62–70]. In line with previous reports supporting that NCL phosphorylation at different sites regulates its interaction with target mRNAs in response to various cellular stresses[55,71], we also showed that NCL dephosphorylation and sumoylation only occurs at later times following LPS treatment that coincide with nucleolar fusion (Figs. 1, 7). This implies that PTM dynamics of NCL at multiple sites or various times following infection

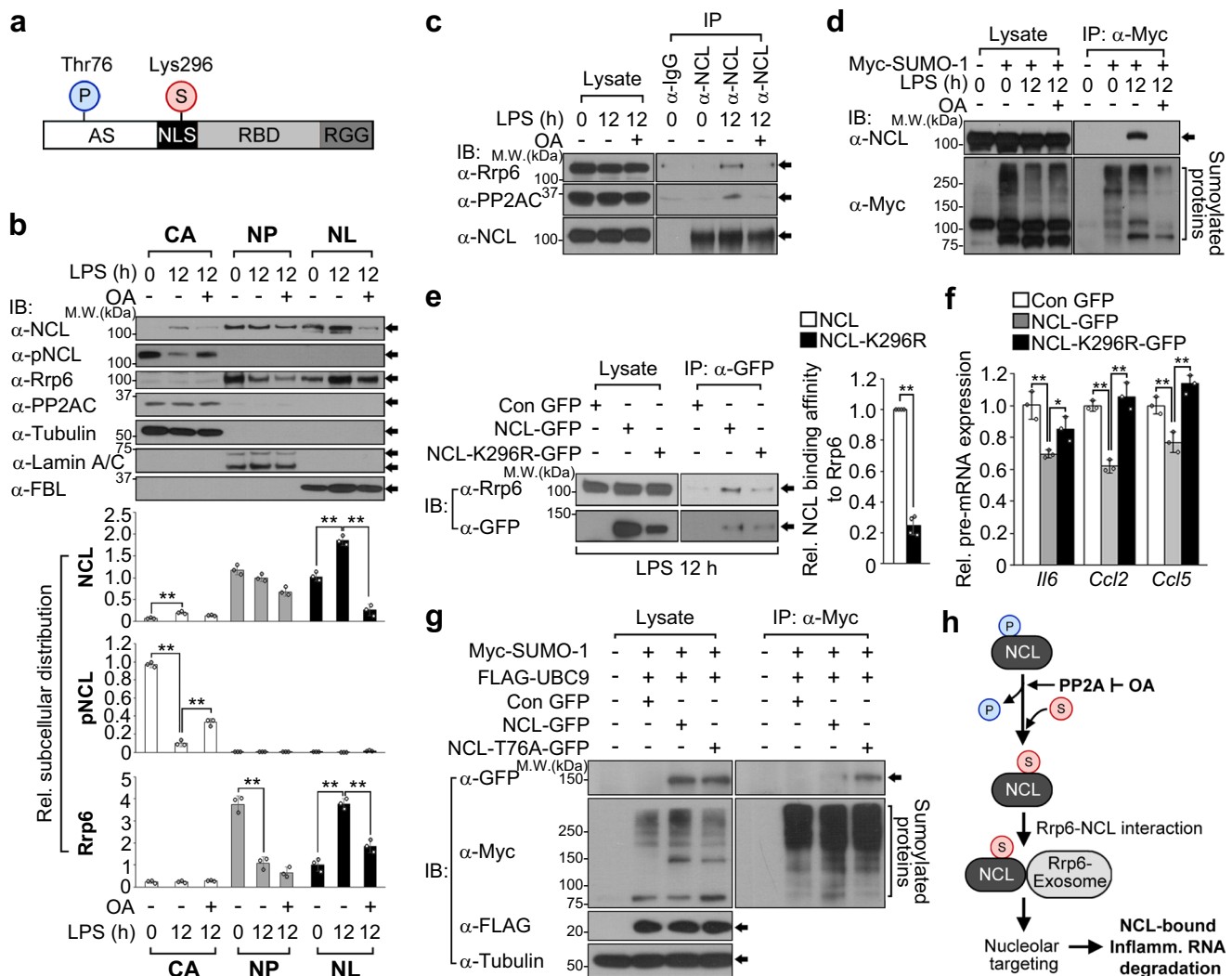

**Fig. 7 | Dynamics of NCL PTMs determines its functional activity during infection. a** Domain architecture and PTM sites of mouse NCL. AS acidic stretches, NLS nuclear localization signal, RBD RNA-binding domain, RGG arginine/glycine-rich region, P phosphorylation, S sumoylation. **b** Subcellular distribution of NCL and Rrp6 or NCL phosphorylation status in 12 h LPS-stimulated RAW 264.7 cells with or without 20 nM OA treatment for 12 h. All bands are quantified in bottom graphs. **c, d** Effect of LPS or OA on Rrp6-NCL or Rrp6-PP2A interaction (**c**) or NCL sumoylation (**d**) in LPS-stimulated RAW 264.7 cells. **e** Influence of GFP-tagged wild-type or mutant NCL (NCL-GFP or NCL-K296R-GFP) on Rrp6 binding ability in wild-type NCL- or NCL-K296R-GFP-expressing MEFs under LPS stimulation. **f** Comparison of inflammatory pre-mRNA expression levels between NCL-GFP and NCL-K296R-GFP-expressing MEFs under 12 h LPS stimulation. **g** Analysis of NCL sumoylation in Myc-SUMO−1- and FLAG-UBC9-expressing MEFs that transiently expressed wild-type NCL- or NCL-T76A-GFP under normal conditions. **h** A model illustrating functional crosstalk between NCL phosphorylation and sumoylation in response to LPS. LPS-induced PP2A-mediated NCL dephosphorylation is critical for its subsequent sumoylation, resulting in nucleolar targeting of Rrp6 for NCL-bound inflammatory pre-mRNA decay. Arrows represent the corresponding proteins immunoblotted by the indicated antibodies. *P* values are determined by unpaired two-tailed *t* test. *$P < 0.05$; **$P < 0.01$ (Student's *t* test). All data are representative of three independent experiments and are presented as means ± s.d. in (**b**, **e**, **f**). Source data are provided as a Source Data file.

might be involved in mediating a diversity of subcellular localizations, RNA-binding specificities, or functions (e.g., stability vs. decay) to produce the maximum defensive response against infection.

Interestingly, we found that nucleoli develop fused morphologies along with increased size in later stages of infection (Fig. 1a, b). Since mitochondrial fusion is selectively induced by increased energy demand and stress, thus maximizing its functional ability[72–74], it is possible that the capacity of nucleolar fusion might be correlated with the maximal ability of inflammatory pre-mRNA degradation. In the same manner, bacterial infection may evoke microtubule rearrangement-mediated nucleolar fusion at later times, possibly maximizing the effectiveness of inflammatory pre-mRNA decay in preventing hyperinflammation and maintaining immune homeostasis. Because the LPS- or CpG-DNA-evoked TLR signaling pathway is likely to influence microtubule organization[75–77], it may provide a starting

point for nucleolar fusion. In addition, macrophages are well known to be phagocytic cells with cytoskeletal dynamics that, compared with those of fibroblasts, are especially suited for active circular ruffling and delivery of extracellular components[78]. Thus, LPS-mediated nucleolar fusion and Rrp6 nucleolar translocation may be correlated with the differences in cytoskeletal dynamics between macrophages and other cells. In favor of this possibility, nucleolar translocation of Rrp6-exosome components in response to LPS treatment only occurred in RAW 264.7 macrophages and was not observed in HeLa or NIH3T3 cells (Supplementary Fig. 8e). Further studies are needed to better understand how the dynamics of nucleolar fusion can maximize the capacity for inflammatory RNA decay under infectious conditions.

Our findings suggest that NCL specifically binds to the C/U-rich sequences positioned in intronic regions, which is responsible for nucleolar targeting and instability of inflammatory pre-mRNAs (Fig. 5).

The C/U-rich sequences recognized by NCL look similar to the poly-pyrimidine tract, which is especially rich with uracil and typically shorter than 20 nucleotides in length[79,80]. In addition, these sequences are often located 5-40 base pairs before the 3′-end of the intron being spliced and promotes association with various RNA-binding proteins, such as hnRNP, U2AF, or polypyrimidine tract binding protein (PTB), leading to RNA splicing[81–83]. However, unlike the polypyrimidine tract, the C/U-rich motif of inflammatory RNAs contains nearly an even number of cytosines and uracils, is over 20 nucleotides in length, and located at various intronic positions (Fig. 5b). Indeed, we did not observe any defects in inflammatory RNA splicing in *Ncl*-knockdown cells (Fig. 4b and Supplementary Fig. 5a). Nevertheless, it remains possible that NCL could bind to the polypyrimidine tract, regulating inflammatory RNA splicing or stability, positively or negatively, through association or competition with U2AF and PTB. Interestingly, we found that *Ccl4* pre-mRNA also accumulated in the nucleolus, and its degradation was significantly decreased in *Rrp6*-depleted RAW 264.7 cells, but not in *Ncl*-depleted cells. These results indicate that another delivery protein may act as a translocation guide by associating with the target substrate and recruiting the Rrp6-exosome complex to the nucleolus for its selective degradation. It has long been suggested that intron density plays an important role in governing alternative splicing, thus raising the probability of exon creation with increasing intron length[84–86]. Consistent with these studies, we showed that the intron to exon ratio of inflammatory genes is much higher than that of non-inflammatory genes which are essential for maintaining immune homeostasis. This implies the intron may be an important causal factor that determines functional diversity in response to different environmental conditions, and might also explain why primate evolution is more prevalent at longer intron lengths. In addition, nucleolar enrichment of intron-containing protein-coding RNAs indicate the nucleolus might be involved in different biological functions beyond the currently known functions.

Based on our findings, autoimmune diseases, septic reaction, or cancers may have defects in nucleolar function, and the down-regulation or mutation of NCL or the Rrp6-exosome complex may contribute to these conditions. Such insights into the molecular pathology of inflammation-associated diseases might serve as a way to restore immune homeostasis and contribute to the development of improved therapies.

## Methods

### Plasmids, antibodies, and reagents
Mouse NCL constructs were subcloned into a pEGFP-N3 (Clontech), pMSCV (Clontech), and pGEX-6P-1 (Cytiva) vectors. Mouse *Fbl*, *Il1b*, *Il1b* ΔC/U-rich, *Cox6a2*, *Rrp43*, *Rrp46*, *Ubc9*, and *Sumo1* were subcloned into the pMSCV vector from mouse cDNA library or genomic DNA. pCI-FLAG-*mRrp6* was a gift from Zissimos Mourelatos (Addgene plasmid #60045). NCL-T76A-GFP and NCL-K296R-GFP were generated by site-directed mutagenesis. The small hairpin RNA (shRNA) oligonucleotides against *Gfp*, *Ncl*, 3′UTR of *Ncl*, *Pum3*, *Pno1*, *Nol6*, *Npm1*, *Rrp6*, *Rrp43*, *Rrp46*, or *Ppp2cb* were annealed and inserted into modified pSUPER retroviral vector (Oligoengine) containing the U6 promoter. All constructs were verified by DNA sequencing. Information about antibodies, reagents, and primers is listed in Supplementary Data 3.

### Mice and cell lines
C57BL/6N mice (4–8 weeks) were purchased from the OrientBio. All mice were maintained in the specific pathogen-free facility of the Yonsei Laboratory Animal Research Center at Yonsei University according to Korean Food and Drug Administration guidelines. All animal experiments were reviewed and approved by the Institutional Animal Care and Use Committee of the Yonsei University (YLARC, No. IACUC-A-201712-467-02). Mouse RAW 264.7 macrophages (TIB-71, ATCC), mouse embryonic fibroblasts (MEFs) (CRL-2907, ATCC), human embryonic

kidney 293T cells (HEK293T) (CRL-11268, ATCC), and HeLa cells (CCL-2, ATCC) were cultured in Dulbecco's Modified Eagle's Medium (DMEM) supplemented with penicillin-streptomycin (HyClone) and 10% heat-inactivated fetal bovine serum (HyClone). Mouse NIH3T3 fibroblasts (CRL-1658, ATCC) were cultured in DMEM with 10% bovine calf serum (HyClone). Cells were grown at 37 °C in humidified air with 5% $CO_2$. For generation of bone marrow-derived macrophages (BMDMs) or dendritic cells (BMDCs), mouse bone marrow cells were obtained from the femurs and tibiae of wild-type C57BL/6N mice at 6–8 weeks old. BMDMs or BMDCs were cultured in medium containing macrophage colony-stimulating factor (25 ng ml$^{-1}$, BioLegend) or granulocyte macrophage colony-stimulating factor (10 ng ml$^{-1}$; BioLegend), respectively. Fresh medium was replenished every 2 or 3 days. BMDMs or BMDCs were fully differentiated after 7 days of culture.

### Transfection and retroviral transduction
RAW 264.7 macrophages, MEFs, and HEK293T cells were transfected with the indicated plasmids using Lipofectamine 3000 (Invitrogen) or OmicsFect (Omics Biotechnology) in serum-free and antibiotic-free DMEM for 24–36 h. For producing retroviral particles, HEK293T cells were transfected with plasmids encoding VSV-G and Gag-Pol, as well as the target construct subcloned into a retroviral vector. After 36–48 h post-transfection, supernatants containing viral particles were filtered through a 0.45 μm syringe filter. Cells were transduced with retroviruses and then incubated with polybrene (4 μg ml$^{-1}$) for 4 h to achieve maximum knockdown efficiency, followed by selection in medium with puromycin (6 μg ml$^{-1}$).

### Immunofluorescence assay (IFA)
For immunofluorescent staining, cells were fixed with 3.7% formaldehyde for 10 min and permeabilized with 0.2% Triton X-100 for 10 min. After blocking with 2% bovine serum albumin (BSA) in PBS (PBA) for 30 min, the samples were incubated with the appropriate primary antibody in 2% PBA at room temperature for 1 h or at 4 °C for overnight. Bound antibody was visualized with an Alexa Fluor 488- or Alexa Fluor 568-conjugated secondary antibody (Invitrogen). DAPI (4′, 6-diamidino-2-phenylindole, Sigma-Aldrich) was used as a nuclear counterstain. All images were collected using a LSM 900 confocal microscope (Carl Zeiss) and the nucleolar size and the intensity profiles of each cell on the microscopic images were analyzed by the ZEN blue software. For details on measurement of the volume of nucleoli, we calculated nucleolar volume assuming an ellipsoid shape, $V = 4\pi abc/3$, where a, b, and c are the lengths of all three semi-axes of the ellipsoid. 3D image stacks reconstructed from a 2D stack of confocal images by Zen blue were used to measure each length of ellipsoid. To quantify the fluorescence of intensity of pre-mRNA foci (RNA-FISH) and nucleolar enrichment (Rrp proteins), different fluorescence signal bandwidth images were combined and then the fluorescence intensity profile in the cross-sectional lines were plotted as a function of X−Y distance across the cell.

### Live cell microscopy
RAW 264.7 cells that stably expressing FBL-GFP were seeded into 35-mm glass-bottom culture dishes (SPL Life Science) and maintained in a microscope stage incubator at 37 °C in humidified air with 5% $CO_2$ throughout the experiment. Confocal microscopy was performed using Zeiss LSM 900 confocal microscope with a ×40/1.2 numerical aperture (NA) water immersion objective. After LPS treatment, images were acquired at 10 to 15 min intervals for 15 h using DIC/GFP channels.

### Bacterial infection and recovery model
*E. coli* strain DH5α was used for in vitro infection and were grown in LB broth at 37 °C overnight. RAW 264.7 cells cultured in 10% FBS supplemented DMEM without penicillin and streptomycin were infected by 50 multiplicity of infection (MOI) bacteria diluted in serum-free

DMEM. Infected cells were further incubated at 37 °C for 2 h after centrifuging at $600 \times g$ for 5 min and then washed with phosphate-buffered saline (PBS) followed by supplementing the complete cell culture medium. For cell recovery from the resolution of inflammation, 12 h LPS-stimulated RAW 264.7 cells were transferred into a new plate after washing with PBS twice and incubated with fresh media for 24 h.

## Nucleolar fractionation

Nucleoli were isolated from LPS-stimulated RAW 264.7 cells ($2 \times 10^7$) as previously described[31]. The cells were suspended with 3 ml cold hypotonic buffer (10 mM HEPES, pH 7.9, 1.5 mM MgCl₂, 10 mM KCl, 0.5 mM dithiothreitol [DTT], 0.05% nonyl phenoxypolyethoxylethanol [NP-40], and protease inhibitor cocktail [PIC]) on ice for 10 min, vortexed vigorously for 10 s, and centrifuged at $218 \times g$ for 5 min. The nuclear pellets were suspended in 3 ml S1 solution (0.25 M sucrose, 10 mM MgCl₂, and PIC) and layered over 3 ml S2 solution (0.35 M sucrose, 0.5 mM MgCl₂, and PIC) followed by centrifugation at $1430 \times g$ for 5 min. Purified nuclei were suspended in 3 ml S2 solution and sonicated on ice for 10 s pulses with 15 s intervals between each pulse for 1 min using a VCX 500 sonicator with a 3-mm tapered microtip (Sonics and Materials) at 40% amplitude. Extracted nucleoli were layered over 3 ml S3 solution (0.88 M sucrose, 0.5 mM MgCl₂, and PIC) and centrifuged at $3000 \times g$ at 4 °C for 10 min. Isolated nucleoli were washed in 500 µl S2 solution followed by a centrifugal force of $1430 \times g$ at 4 °C for 5 min.

## RNA-sequencing (RNA-seq) analysis

Total RNAs were isolated using TRIzol reagent (Invitrogen) from cytoplasmic, nucleoplasmic, and nucleolar fractions in LPS-stimulated RAW 264.7 cells for the indicated time points. To exclude potential DNA contamination, the isolated RNAs were cleaned up using RNeasy mini kit (Qiagen) and validated for RNA integrity via running on 1% agarose gels in Tris-borate-EDTA buffer[87]. For Illumina sequencing, rRNAs were removed from total RNAs twice with a Ribo-zero rRNA removal kit (Illumina) and then synthesized cDNA libraries according to the manufacturer's instructions (TruSeq Stranded Total RNA Library Prep kit, Illumina). PCR products of about 200 base pair (bp) were excised for size selection. After determining the concentration and quality of products by Bioanalyzer, RNA-seq was performed by Illumina HiSeq2000 using 50-bp paired-end readings. Raw data were analyzed by the FastQC tool for quality control (http://www.bioinformatics.babraham.ac.uk/projects/fastqc) and mapped to the Mus musculus reference genome (mm10) using Tophat2 (v.2.1.1) or Bowtie2 (v.2.2.9) with default parameter settings. For visualization of aligned reads in subcellular fractions by Integrative Genomics Viewer (IGV) v.2.3.93[88], the sequencing data in bam format were converted to sequencing coverage tracks in bigwig format normalized to 1× depth of coverage as reads per genomic content using deeptools bamCoverage[89]. To evaluate the RNA abundance and distribution of inflammatory or non-inflammatory genes across different samples of each fraction, RNA-seq data was subjected to StringTie to calculate TPM values from sorted Bam files and then to exon-intron split analysis to quantify pre-mRNA levels. Finally, the datasets applied quantile normalization to compare between subcellular fractions[90–92]. Among total protein-coding genes ($n = 6928$) yielded from the nucleoplasmic fractions at each time point after LPS stimulation, we selected protein-coding genes ($n = 5097$) expressed above 5 TPM value with the exception of anomalous genes with higher intronic read counts in the cytoplasm. We further sorted protein-coding genes ($n = 5097$) into non-inducible (3636 genes, $0 < FC \leq 1$), moderately inducible (1023 genes, $1 < FC \leq 2$), and highly inducible (438 genes, $FC > 2$) categories. Inflammatory genes (115 genes) were further sorted from highly inducible genes based on immune-related gene ontology terms (GO:0006955, GO:0002376, GO:0006952, and GO:0034097). To quantify nucleolar intronic enrichment of specific genes, the intronic TPM in nucleolar fractions was divided by that in nucleoplasmic

fractions at 0, 2, 6, 12, or 15 h LPS stimulation except non-stimulation as LPS-inducible genes were barely expressed under normal conditions ([NL + 1]/[NP + 1]). Heat map was generated by multiple experiment viewer (v.4.9.0) with the log-transformed intron/exon ratio of inflammatory or non-inflammatory genes in nucleoplasmic fraction[93].

## RNA fluorescence in situ hybridization (RNA-FISH)

RNA-FISH probes designed from LGC Biosearch Technology's Stellaris Designer program (https://www.biosearchtech.com/stellaris-designer). LPS-stimulated RAW 264.7 cells were fixed with 3.7% formaldehyde and permeabilized with 70% ethanol at 4 °C for at least 1 h. RNA probes were incubated with hybridization buffer (10% formamide, 2× saline sodium citrate [SSC], and 100 mg ml⁻¹ dextran sulfate) in a dark humidified incubator at 37 °C for 4 h. Unbound probe was removed with washing buffer (10% formamide, 2× SSC, and DAPI) and samples were mounted with the antifade reagent. For analyzing the spatial preferences of inflammatory pre-mRNA foci with RNA polymerase II (RNA-pol II), nucleoli, or InSAC, the pre-mRNA FISH signals with immunofluorescence for anti-pS2, FBL-GFP/anti-FBL, or GFP-TDP-43, respectively, in hybridization buffer. Subcellular distributions were visualized by Alexa Fluor 647-conjugated secondary antibody (Invitrogen) and the fluorescence intensity and size of pre-mRNA foci were quantified by ZEN blue software. For quantitative measurement of colocalization, two cross-sections through the center of each cell were selected, and the red (inflammatory or non-inflammatory pre-mRNAs) and green (nucleolus) signal intensities in the cross-sectional lines were plotted as a function of X–Y distance across the cell. The probe sequence and antibody information are listed in Supplementary Data 3.

## RT-PCR and RT-qPCR

Total RNAs were extracted from nuclei or from mouse lung- or liver-resident cells lacking *Ncl* using RNA-prep kit (GeneAll) or TRIzol reagent (Invitrogen) according to the manufacturers' instructions. Complementary DNA (cDNA) was synthesized from 2.0 µg RNA using random hexamers, oligo (dT), and reverse transcriptase (Enzynomics) at 42 °C for 1 h. The PCR reaction was generally performed with gene-specific primers under the following cycling conditions: 25 to 35 cycles of 95 °C for 20 s, proper annealing temperature for 20 s, and 72 °C for 30 s. PCR products were detected in agarose gels containing ethidium bromide. The qPCR reaction was performed with the Applied Biosystems QuantStudio3 Real-Time PCR System (Life Technologies). Relative values were quantified by the change-in-threshold method compared with control, and glyceraldehyde 3-phosphate dehydrogenase (*Gapdh*) mRNA was used for normalization. For measuring the relative nucleolar pre-mRNA levels of *Myc-Il1b*, *Myc-Il1b* ΔC/U-rich and *Myc-Cox6a2*, total RNAs were isolated from the nucleoplasm and nucleoli in unstimulated or LPS-stimulated RAW 264.7 cells, and the percentage of pre-mRNAs were calculated by dividing nucleolar pre-mRNA levels by the sum of nucleoplasmic and nucleolar pre-mRNA levels. All primer sequences and product sizes are listed in Supplementary Data 3.

## Northern blot analysis

Total RNAs isolated from nuclei were resolved in 50% formamide, boiled at 95 °C for 5 min, and separated on 1.8% formaldehyde agarose gel in MOPS (3-[N-morpholino] propanesulfonic acid) buffer. By utilizing downward alkaline capillary transfer, RNAs were transferred to positively charged nylon membrane (Amersham Hybond-XL) with alkaline transfer buffer (0.01 N NaOH and 3 M NaCl) for 4 h. The membrane was prehybridized with ultrasensitive hybridization buffer (Invitrogen) for 30 min and the 20 pmol of [γ-³²P]-labeled DNA probes were applied for hybridization of blots at 42 °C for overnight. Membrane was washed three times with 2× sodium chloride-sodium phosphate-EDTA (SSPE) containing 0.1% sodium dodecyl sulfate (SDS) and

hybridization signals were detected with Typhoon Biomolecular imager. The band intensity was quantified using densitometry and normalized by 18S rRNA. The probe sequence information is listed in Supplementary Data 3.

### RNA decay assay

RAW 264.7 cells were stimulated with LPS for 12 h, followed by treating 6 µg ml$^{-1}$ Act.D, and then were further incubated for the indicated time points. Total RNAs were extracted from the nuclear, nucleoplasmic, or nucleolar fractions of RAW 264.7 cells and then quantified by qPCR analysis with the specific primer pairs. All primer sequences are listed in Supplementary Data 3.

### Luciferase assay

RAW 264.7 cells that stably expressing NCL-GFP or shRNA against *Gfp*, *Npm1*, or *Ncl* were seeded to a 12-well plate and transfected with the Renilla reporter and the *Il6* promoter luciferase reporter as previously described[94]. After 24 h, cells were stimulated with LPS alone or along with etoposide (10 µM) for 12 h and lysed with lysis buffer according to manufacturer's instructions (Promega). The luciferase activity was determined by using the Dual-luciferase reporter assay system to measure firefly luciferase activity, which was normalized to Renilla luciferase activity.

### In vivo RNA interference experiment

*Ncl* depletion in mouse lung- or liver-resident cells was mediated by siRNA against NCL with in vivo-jetpei (Polyplus) or Invivofectamine 3.0 (Invitrogen), respectively. According to the manufacturer's instructions, the siRNA-in vivo-jetpei complex (120 µg siRNA) was injected twice or siRNA-Invivofectamine complex (50 µg siRNA) was injected intravenously into the tail veins of 4-week-old C57BL/6N mice. Two days later, LPS (300 µg kg$^{-1}$) was intravenously injected to the tail veins of mice for 10 h, and then the level of inflammatory RNAs, proteins, or *Ncl* knockdown was evaluated by qPCR or ELISA. We monitored control or lung-specific *Ncl*-depleted mice survival every 4 h for 2 days after intravenous administration of PBS or LPS (6 mg kg$^{-1}$).

### Enzyme-linked immunosorbent assay (ELISA)

RAW 264.7 cells expressing shRNA against *Gfp*, *Ncl* or *Rrp6* were stimulated with LPS for 12 h. IL-1β, IL-6, CCL4, and CCL5 levels in the culture supernatant were measured by ELISA according to the manufacturer's instructions (R&D Systems). After mice were challenged by LPS for 10 h, blood was obtained from the retro-orbital plexus with the use of heparinized micro-hematocrit tubes in deep anesthesia. Heparin-anticoagulated plasma was used to perform ELISA.

### PAR-CLIP analysis

For identifying NCL-binding consensus motif(s), PAR-CLIP was performed as described previously[41,42]. $1 \times 10^8$ RAW 264.7 cells were incubated in medium supplemented with 100 µM 4-thiouridine (4SU) for 16 h, in the presence of LPS for 12 h. The cells were washed with PBS, and irradiated with 0.15 mJ cm$^{-2}$ and 365 nm ultraviolet (UV) light in a UV crosslinker (Spectrolinker XL-1500) to crosslink the labeled RNA to RNA-binding proteins. After isolation of nucleoli from cell lysates, nucleolar extracts were treated with 1 U ml$^{-1}$ RNase T1 (Fermentas) and NCL proteins were immunoprecipitated with anti-NCL antibodies and Protein A/G-sepharose beads. The immunoprecipitates were further digested with 100 U ml$^{-1}$ RNase T1 and the beads were washed in lysis buffer and suspended in one bead volume of dephosphorylation buffer. RNAs were dephosphorylated and radioactively labeled with [γ -$^{32}$P]-ATP. The protein-RNA complexes were separated by SDS-PAGE and visualized by autoradiography. The radioactive bands migrated as a 110 kDa were eluted from the gel and all proteins in eluted bands were removed by digestion with 0.2 mg ml$^{-1}$ proteinase K. The RNAs were isolated

by acidic phenol/chloroform extraction and ethanol precipitation, converted into a cDNA library, and sequenced using an Illumina platform (HiSeq2500). Processed reads were aligned to the reference genome (mm10) by the Bowtie algorithm, allowing for two alignment errors (mutation, insertion, or deletion). For each read, only the best mapping was reported out of a maximum of 10 genomic matches. After the conversion subtraction, reads that mapped to only one genomic location were retained for further analysis.

The PAR-CLIP data were analyzed by using the PARpipe analysis, which is a pipeline wrapper for PARalyzer (v.1.5) (https://ohlerlab.mdc-berlin.de/software/PARpipe_119/). Raw sequencing reads were aligned to the mouse genome (mm10) and overlapped, and uniquely aligned reads were assembled into groups. Those groups with greater read kernel density estimates than the background, a read depth of at least three, and at least two independent diagnostic T-to-C mutations indicating crosslinking were considered high-confidence binding sites or clusters and annotated using the following database (genco-de.v24.chr_patch_hapl_scaff_fixed.annotation.gtf). To identify the NCL-binding motifs, MEME-ChIP (v.5.0.4) was used with 4899 NCL-crosslinked sequence reads in intronic regions from PAR-CLIP library (Supplementary Data 2). 106 NCL target genes containing the C/U-rich motif in their introns were sorted from LPS-inducible genes (438 genes) based on the PAR-CLIP dataset (Supplementary Data 1). The problem of PCR amplification bias was avoided by limiting the number of PCR cycles.

### RNA immunoprecipitation (RIP) assay

LPS-stimulated RAW 264.7 cells were washed with cold PBS twice and then fractionated by hypotonic buffer containing PIC and recombinant RNase inhibitor (TaKaRa). Isolated nuclei were suspended in RIP buffer (50 mM Tris-HCl pH 7.4, 0.05% NP-40, 100 mM NaCl, RNase inhibitor, and PIC) and sonicated for 2 s with 3 s intervals 10 times at 20% amplitude. Nuclear extracts were incubated with anti-NCL antibodies for 5 h, followed by protein G-Sepharose beads (Sigma-Aldrich) for 1 h and the beads were washed three times with RIP buffer. NCL-bound RNAs were isolated by RNA purification kit and detected by RT-PCR or qPCR assay with the indicated primers described in Supplementary Data 3. For evaluating the NCL-binding affinity for the *Myc*-tagged minigenes, nuclear extracts from LPS-stimulated RAW 264.7 cells were incubated with anti-NCL antibodies. Purified NCL-bound RNAs were detected by qPCR with the primers recognizing *Myc* and intronic regions of *Il1b* or *Cox6a2*. Relative values were quantified by the change-in-threshold method compared with control, and *Gapdh* pre-mRNA was used for normalization.

### Recombinant protein purification

*E. coli* strain BL21 (DE3) competent cells were transformed with plasmids encoding GST-tagged NCLΔAS and grown in LB medium to an OD600 of 0.6–0.8 at 37 °C. Protein production was induced by addition of 0.5 mM isopropyl-β-D-1-thiogalactopyranoside (IPTG) and cells were further grown at 37 °C for 3 h. Collected bacterial pellets were disrupted by the sonicator for 2 s pulses with 3 s intervals in ice-cold lysis buffer (20 mM Tris-HCl pH 7.4, 300 mM NaCl, 0.1% NP-40, 20 mM EDTA, and 10% glycerol) and centrifuged at $15,928 \times g$ for 10 min. The supernatant was incubated with glutathione (GSH) agarose beads (Elpis Biotech) for 3 h, and non-specific proteins were cleared with lysis buffer 5 times. Proteins were eluted with elution buffer (50 mM Tris-HCl pH 8.0 and 20 mM GSH). Eluted proteins were desalted by PD MidiTrap G-25 column (Cytiva), concentrated by Amicon Ultra-4 centrifugal filter unit (Sigma-Aldrich), and stored in storage buffer (25 mM Tris-HCl pH 7.4, 100 mM glycine, and 10% glycerol). The size and concentration of purified proteins were confirmed by the Bradford protein assay (Bio-Rad) and Coomassie blue staining.

## RNA electrophoretic mobility shift assay (EMSA)

All RNA probes were labeled with [γ-$^{32}$P]-ATP (1 μCi) by T4 Polynucleotide kinases (Fermentas) at 37 °C for 1 h, and T4 Polynucleotide kinases and free ATP were removed by QIAquick nucleotide removal kit (Qiagen). $^{32}$P-labeled probes (0.5 nM) were incubated with concentration gradients of recombinant NCL (0–5 μM) in reaction buffer (20 mM Tris-HCl pH 7.4, 150 mM NaCl, 4 mM MgCl$_2$, 5% glycerol, 6 mM DTT, 0.1% NP-40, and RNase inhibitor) at room temperature for 20 min. Sample mixtures were separated by native PAGE using 12% (bottom)/6% (top) step-gradient gels at 150 V for 45 min and dried at 60 °C for 1 h. The radioactive signals were detected using the Amersham Typhoon biomolecular imager and quantified using densitometry. Dissociation constants ($K_d$) and curves were analyzed from the band intensity of free RNA and bound RNA by Prism software.

## Anisotropy analysis

Binding of GST-NCL to 3′-fluorescein (Fl)-tagged *Il1b*-[C/U]$_{21}$ or *Il6*-[C/U]$_{20}$ probes was evaluated by changes in the fluorescence anisotropy of the substrates as a function of protein concentration. Limiting concentrations of Fl-tagged substrates (20 nM) were incubated with varying concentrations of purified NCL in 20 mM Tris-HCl (pH 7.4), 150 mM NaCl, 4 mM MgCl$_2$, 10 mM DTT, 1% NP-40, 0.5% glycerol, and 1 U/μl RiboLock RNase inhibitor. The assay was performed in 96-well non-binding black plates (Corning), with fluorescence polarization measured using a Spectramax iD5 microplate reader (Molecular Device) with the excitation and emission wavelengths 490 nm and 530 nm, respectively. Milli-polarization units (mP) were used to express fluorescence polarization values defined by the equation $mP = 1000 \times ([I_{\parallel} - GI_{\perp}]/[I_{\parallel} + GI_{\perp}])$, where $I_{\parallel}$ and $I_{\perp}$ are parallel and perpendicular emission intensity measurements. The data was corrected for background (RNA sample alone), and G-factor, then fitted by a one-site binding model using the Equation, $y = y_0 + (y_{max} * x/K_d + x)$, where $y$ is the corrected fluorescence polarization, $y_0$ is polarization value without NCL, $y_{max}$ is maximum binding fluorescence polarization signal, $x$ is the purified NCL concentration, and $K_d$ is the dissociation constant. $x$ and $y_{max}$ were used as fitting parameters and nonlinear regression was performed using Prism. Measurements were taken after 5 min incubation between NCL and RNA at room temperature.

## Co-immunoprecipitation (Co-IP) and immunoblot analysis

Cells were stimulated with LPS in the absence or presence of OA for 12 h and lysed with 1% NP-40 (for the interaction of NCL with Rrp6 or PP2AC) or 1% digitonin (for the interaction of NCL with Rrp43 or Rrp46) with PIC. The lysates were incubated with primary antibodies followed by protein G-Sepharose beads at 4 °C and the beads were washed twice with 0.1% NP-40 or 0.1% digitonin. Proteins eluted by boiling in denaturing buffer (50 mM Tris-HCl, pH 6.8, 2% SDS, and 5% β-mercaptoethanol) were separated by SDS-PAGE, transferred to polyvinylidene fluoride (PVDF) membranes (Millipore), and probed with the primary antibodies. Proteins were detected using HRP-conjugated secondary antibodies and enhanced chemiluminescence (Advansta). For details on NCL sumoylation, we firstly generated RAW 264.7 cells or MEFs cells stably expressed Myc-SUMO1 with or without a SUMO-conjugating enzyme, FLAG-UBC9, to maximize cellular sumoylation levels. The isolated nuclei (RAW 264.7 macrophages) or whole cells (MEFs) were lysed with SUMO lysis buffer (50 mM HEPES, 1% NP-40, 150 mM NaCl, 2.5 mM MgCl$_2$, 0.01% SDS, 0.1 mM EDTA, 0.05% sodium deoxycholate, 10 mM N-Ethylmaleimide, and PIC) and sonicated at 20% amplitude for 30 s. Samples were incubated with anti-Myc antibodies followed by protein A-Sepharose (Sigma-Aldrich) at 4 °C and then immunoblot with the indicated antibodies. Antibody concentrations are described in Supplementary Data 3.

## Liquid chromatography-tandem mass spectrometry (LC-MS/MS)

After Co-IP with NCLΔAS-GFP-expressing RAW 264.7 cells stimulated with LPS for 12 h, eluted samples were separated by 8% SDS-PAGE and visualized by silver staining. The bands of interest were excised from the gel and polypeptides were digested by trypsin proteases in 50 mM ammonium bicarbonate on ice for 45 min followed by incubation at 37 °C for 12 h. Tryptic peptides were analyzed by Agilent 6530 Accurate-Mass Q-TOF with a nanochip column (Agilent, 150 mm × 0.075 mm) as previously described[95]. MASCOT (Matrix Science, London, U.K.; v.2.2.04) was used to identify peptide sequences in the protein sequence database (nr_mouse). Database search parameters for LC-MS/MS were as follows: fixed modification, carboxyamidomethylated at cysteine residues; variable modification, oxidized at methionine residues; maximum allowed missed cleavage, 1; peptide MS tolerance, 100 ppm; and fragment MS tolerance (LC-MS/MS), 0.1 Da. Individual ion scores showing greater than 42 for LC-MS/MS were regarded as significant ($P < 0.05$).

## Nuclear and cytoplasmic fractionation

MEFs were transiently transfected with plasmids encoding control GFP, wild-type NCL-GFP, or NCL-K296R-GFP for 24 h followed by 12 h LPS stimulation and the cytoplasm from these cells was fractionated using hypotonic buffer. Nuclear pellets were rinsed with hypotonic buffer, suspended in nuclear lysis buffer (50 mM Tris-HCl pH 7.4, 0.05% NP-40, 100 mM NaCl, and PIC), and sonicated for 5 cycles of 1 s pulses with 3 s intervals at 20% amplitude on ice.

## Statistical analysis

All experiments were repeated at least three times with consistent results. Data are presented as means, standard deviation (s.d.), and standard error of mean (s.e.m.) as noted in the figure legends. Statistical analyses were performed using Graphpad Prism 8.0 or Microsoft Excel 2016 software. Statistical differences between two means were evaluated with the two-tailed, unpaired Student's $t$ test, Mann–Whitney $U$ Test, or log-rank test. Differences with $P$ values below 0.05 were considered significant. No samples were excluded from the analysis. The data were normally distributed, and the variance was similar between groups. No statistical method was used to predetermine sample sizes. Instead, sample size determinations were based on previous experiences with experimental variability. The investigators were not blinded to allocation during experiments or outcome assessments.

## Reporting summary

Further information on research design is available in the Nature Research Reporting Summary linked to this article.

## Data availability

The RNA-seq and PAR-CLIP data have been deposited in NCBI's Gene Expression Omnibus (GEO) and are accessible through GEO Series accession number GSE159346 and GSE211163, respectively. The Mass spectrometry data are available in the ProteomeXchange Consortium via the jPOST partner repository with the accession code identifier PXD030716. The remaining data are available within this published article and its Supplementary Information files. Source data are provided with this paper.

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

## Acknowledgements

We thank Dr. Victoria Stepanova (Pathology and Laboratory Medicine Perelman School of Medicine at the University of Pennsylvania) for kindly providing full-length wild-type mouse NCL plasmid. We appreciate Junghyun Cho, Mikhail Fomin, Kelly Misare, Dimitrios G Anastasakis, Duncan Claypool, and Amir Manzourolajdad for providing technical support of PAR-CLIP. This work was supported by the Samsung Science & Technology Foundation (SSTF-BA1801-08) and the National Research Foundation of Korea (NRF) funded by the Ministry of Science, ICT, and future planning (NRF-2016R1A5A1010764, NRF-2017R1E1A1A01074135, NRF-2022R1A2C3008614, and NRF-2022M3E5F2018597). S.L. was supported by NRF (NRF-2018R1D1A1B07048930) and the National Cancer Center of Korea (NCC-2010170). J.-H.Y. and K.-W.M. were supported by startup fund from Medical University of South Carolina. A.P. and M.H. were supported by the Intramural Research Program of the National Institute of Arthritis and Musculoskeletal and Skin Diseases. J.E.L. was supported by NRF (NRF-2021R1A2C3004572 and NRF-2021R1A4A2 001389), J.W.L. and K.-W.M. were supported by NRF (NRF-2019R1C1 C1002886 and NRF-2022R1A2C4001528), and T.A.L. was supported by NRF (NRF-2019R1A6A3A01096470 and NRF-2020R1I1A1A01072359). T.A.L., E.A.R., E.L., A.Park, S.K., and J.L.C. were supported by the Brain Korea (BK21) FOUR Program.

## Author contributions

T.A.L. performed most of the experiments. H.H., S.K.C., A.Polash, and J.W.L. performed experiments related to bioinformatics analysis with RNA-seq dataset, northern blot, PAR-CLIP, and anisotropy analysis, respectively. E.A.R., A.Park, E.L., S.K., J.L.C., and J.-H.K. performed biochemical analysis, luciferase reporter assay, cloning, generation of stable expression cell lines, and in vivo experiment. J.E.L., M.H., K.-W.M., S.W.Y., and I.L. provided intellectual input and advised on experimental designs. T.A.L., J.-H.Y., S.L., and B.P. designed experiments and wrote the paper.

## Competing interests

The authors declare no competing interests.
