## [Peer Review File · Nature Communications]

The nucleolus is the site for inflammatory RNA decay during infectionEditorial Note: Parts of this Peer Review File have been redacted as indicated to remove third-party material where no permission to publish could be obtained.

REVIEWER COMMENTS

Reviewer #1 (Remarks to the Author):

In this manuscript, the authors report that nucleolus is an essential site of destabilization of intron-containing inflammatory RNAs during infection. Inflammatory genes have a higher read density of introns than non-inflammatory RNAs, and their pre-mRNAs are highly enriched in the nucleolus during infection. NCL was found to recruit inflammatory pre-mRNAs containing (C/U)-rich sequences and the Rrp6-exosome complex into the nucleolus by direct interaction, thereby acting as a guide factor to enable target RNA delivery to the Rrp6 exosome and subsequent degradation. In addition, depletion of NCL leads to abnormal hyperinflammation and severe lethality in response to LPS. Furthermore, the dynamics of post-translational modification of NCL is what determines its functional activity during the LPS phase. These results represent a novel nucleolus-dependent pathway for maintaining the integrity of inflammatory gene expression and immunological homeostasis during infection.

The study is carefully constructed and clear conclusions are drawn from the individual experiments. The data can be taken as reliable. In order to improve it further, some concerns need to be addressed.

Comments

On page 5, lines 110-111, Fig. 1c is included as an infection experiment, but Fig. 1c is an LPS stimulation experiment, not an infection experiment.

Are alveolar macrophages and Kupffer cells targeted by the lung-specific and liver-specific siRNAs shown in Figure 3c? What are the cells used in Extended Data Figure 4c?

It is interesting that NCL has the similar affinity for C/U-rich sequences as NCL recognition elements (NREs). Is this through the same RBD? Or can they bind in completely different forms? The NRE forms a stem-loop structure; does the C/U-rich sequence form any structure?

Why does Rrp6/43/46 predominantly translocate from the nucleoplasm to the nucleolus and co-localize with NCL in macrophages, even though HeLa and NIH3T3 cells also respond to LPS? (Extended Data Figure 7e)

Is there any evidence that NCL (Thr76) is a direct substrate for dephosphorylation by PP2A?

Authors propose that inflammatory pre-RNAs containing C/U-rich intron sequences are transported to the nucleolus by the NCL-Rrp6 axis, but to what extent is this transport coupled to nucleoli fusion? Does the fused nucleolus then divide (original form) after the resolution of inflammation?

Check the references form (Large/small text, abbreviations).
4,8,10,12,16,31,40,44,45,47,59

Reviewer #2 (Remarks to the Author):

In this manuscript, T.A. Lee et al. propose that the nucleolar localization of inflammatory cytokine pre-mRNAs serves as a mechanism to reduce cytokine expression in stimulated macrophages. This is a very intriguing hypothesis that would establish a novel level of post-transcriptional control that has not been described before. The hypothesis is based on the observation that intron-containing inflammatory cytokine pre-mRNAs are enriched in nucleoli of stimulated macrophages, the description of nucleolin (NCL) as a factor that binds to CU-rich sequences within introns of these pre-mRNA, the finding that knockdown of NCL and other nucleolar proteins causes elevated expression of inflammatory cytokine pre-mRNAs via mRNA stabilization, the discovery that NCL associates with the

nuclear Rrp6-containing exosome, and a set of results indicating that this process is controlled by posttranslational modification (PTM) of NCL.

This mechanism is clearly novel and surprising since nucleoli have so far not been implicated in regulating the expression of inducible genes. In general, I am impressed by the mechanistic insight the authors were able to gain on this novel mechanism. However, I think that several of the connections need to be demonstrated with better experiments or additional controls, and I am not sure that the model put forward is fully congruent with the results presented. Several aspects need to be clarified on this intriguing hypothesis.

Major concerns:

The data in Figure 2a-c indicate that nucleoli are enriched for intact pre-mRNAs since the read distribution seems to correspond to that of genomic DNA (more intronic reads than exonic). What is not clear to me: are these read counts normalized to the length of the underlying sequence? If so, the density of intronic reads would be higher than the density of exonic reads, which would suggest that intronic RNA alone is enriched in nucleoli (sequestration of introns after splicing?). Since it is very hard to grasp the data in Figure 2a-c, I suggest the authors depict gene tracts of some representative inflammatory and non-inflammatory genes with the read density in the three compartments. This will help to clarify which RNA species is exactly enriched in nucleoli.

If pre-mRNAs are enriched in nucleoli, one would presume that they are a) protected from splicing and b) protected from degradation in this compartment. To me, this is difficult to reconcile with the proposed model that NCL causes re-localization of specific pre-mRNAs into nucleoli and targets them for Rrp6/exosome-mediated decay. Maybe the authors could use their fractionation approach to address the stability of different RNA species in the three compartments following transcriptional shut-down with Act.D. (or more elegantly, using a nascent RNA labelling approach to avoid the side effects of Act.D on nuclear organization and homeostasis). They could separately address the stability of pre-mRNA (using intron-exon boundary spanning reads (or amplicons), mRNA (using exon-exon boundary spanning reads or amplicons), and intronic RNA (using intra-intronic reads or amplicons). Obviously, the latter will also report on pre-mRNA. This could help to clarify whether pre-mRNAs are rapidly degraded or stable in nucleoli, and also give information on the decay of intronic RNA after splicing (which is thought to be degraded rapidly in the nucleoplasm). To me, the nature and stability of the observed RNA-pol II transcripts in nucleoli are the main aspects the authors should clarify before postulating a specific model.

If the authors chose to measure nucleolus-associated RNA decay by RNA-Seq, they could also interrogate their data for evidence of 3'-to-5' degradation (i.e. an accumulation of reads towards the 5'end), which would be a good indication for exosome-mediated decay.

Another important point is what the FISH signal of inflammatory pre-mRNAs in (or at the rim of) nucleoli means. Most likely, the observed foci correspond to the sites of transcription - this should be stated in the text. Does this mean that RNA-pol II is active in (or at the edge of) nucleoli? This would be very intriguing. Or is the relocalization of the sites of transcription of inflammatory genes into nucleoli a means of suppressing transcription, given that nucleoli are thought to be sites of privileged RNA-pol I activity (this could fit the idea that inflammatory genes need to be turned off at later stages of macrophage activation). For this question, it would also be important to show where the sites of transcription of control non-inflammatory and non-induced genes are. This aspect might be an important part of the authors' model, and should also be discussed in view of what is known about RNA-pol II in nucleoli.

To me, the discussion is superficial and not very thoughtful. The authors should address which RNA species is targeted to (or synthesized in) nucleoli, and what the physiological meaning of the newly described NCL-Rrp6/exosome decay pathway is. Do their data fit to earlier reports of NCL acting on

mRNA stability? Is what they discovered unique to stimulated macrophages, and could it be involved in the attenuation of inflammatory responses, an important function macrophages fulfill after triggering inflammation?

Specific comments:

Intro line 63: "...the fact only 2% of nuclear pre-mRNAs are converted to mRNAs suggests that incomplete or remnant pre-mRNAs must be actively degraded in the nucleus⁹." This statement is misleading. The study by Brandhorst & McConkey (J Mol Biol, 1974) estimated the turnover rate of pre-mRNA based on radioactive nucleotide incorporation, and essentially reflects the fact that intron are rapidly degraded in the nucleus, while only a small portion of the pre-mRNA sequence (the exons in mRNA) is exported to the cytoplasm and loaded on polysomes. These measurements were global and did not address incomplete or remnant pre-mRNAs.

Figure 1 shows the growth in size and reduced numbers of nucleoli upon stimulation of macrophages: since macrophages become more flat upon activation, the change in cell shape alone could explain the increase in nucleolar area. Hence, the authors should measure the volume of nucleoli.

Fibrillarin is a marker for the dense fibrillar component (DFC) of nucleoli. What happens to the other components: the fibrillar center (FC) and the granular component (GC)?

Line 124, Figure 2a: "approximately 48% of detectable reads in the nucleolus were found to be protein-coding RNAs which exclusively possess introns..." This is an overstatement, the NL fraction still contains about 10% reads from exons. The authors should phrase their observations more carefully, here e.g. that intronic RNA was strongly enriched in the nucleolar fraction. Since exons make up less than 10% of pre-mRNA sequences, this observation is compatible with stable pre-mRNAs being present in NL. This point could be further strengthened by showing the read distribution on gene tracks of a few exemplary inflammatory and non-inflammatory genes (see comment above).

In Figure 2b, the intron/exon ratio in NP of non-inflammatory and inflammatory genes looks similar, whereas Figure 2d shows that this ratio is drastically different between non-inflammatory and inflammatory genes. The data seem to be contradictory. What is also important for this comparison: were the read numbers normalized to the length of the underlying sequence? If not, could it be that the length distribution of introns and exons is different between non-inflammatory and inflammatory genes?

Line 137: "Furthermore, changes in nucleolar inflammatory pre-mRNA levels did mirror that in nucleoplasmic pre-mRNA levels without further reduction of the corresponding cytoplasmic mRNA levels, indicating that these changes might not be due to attenuation of RNA splicing (Extended Data Fig. 2c)." I simply don't understand this sentence, it is way too long and complicated. To me, Extended Data Fig. 2c shows that inflammatory genes are induced in all three compartments, though earlier in NP and NL than in CA. The authors should make an effort to express their ideas in simple terms.

Figure 2e: It seems that the FISH signal of the inflammatory pre-mRNAs is at the edge of nucleoli. Is this seen in all the images? The localization should be quantified over a large number of images. Also, it would be important to show the FISH signal of a few non-inflammatory genes (see also comment above).

Figure 3c: In which tissue was RNA expression measured?

What happens to the inflammatory pre-mRNA foci (i.e. the sites of transcription) after kd of Npm1 or Ncl? This could be tested using e.g. rRNA as a marker of nucleoli, fibrillarin or a component of the exosome.

Figure 4f: How was binding of the reporter mRNA to NCL measured? Is this an RNA-IP approach, or does it include crosslinking? More detail should be given in the text.

Similar to the Myc-minigene mRNA, the authors should test by qPCR if endogenous inflammatory pre-mRNAs are bound to NCL, using a number of non-inflammatory pre-mRNAs as controls. This would help to validate the CLIP data in Figure 4a.

Line 256: Subtitle should say: NCL delivers inflammatory RNAs to the Rrp6-exosome for THEIR degradation

Extended Data Fig. 6d shows that NCL-deltaAS is less active than full-length NCL, especially towards suppressing the expression of IL6 mRNA. Hence, I do not agree with the authors' assertion that "we alternatively used the deletion mutant of NCL lacking acidic stretches (NCLdeltaAS-GFP) as its effects on both nucleolar enrichment and instability of IL1b and IL6 pre-mRNAs were similar to those of full-length NCL in control as well as Ncl-depleted macrophages (line 267).

Figure 5a/b: The finding that NCL appears to interact with Rrp6 only in LPS-stimulated cells is very interesting. However, the authors should also show the control IP (anti-GFP) in LPS-stimulated cells.

As a consequence of this finding, one would predict that kd of NCL causes upregulation of inflammatory mRNAs only in stimulated macrophages, but not under basal conditions. This should be tested systematically for a range of inflammatory mRNAs. Vice-versa, overexpression of NCL should suppress inflammatory mRNAs only in stimulated, but not unstimulated macrophages.

Since NCL kd affects IL1b and IL6, but not Ccl4 mRNA expression (Figure 3c), whereas Rrp6 overexpression reduces the expression of all three mRNAs (Extended Data Figure 7b), I am concerned about the specificity of the NCL-exosome connection. Maybe Rrp6 overexpression is not a good approach since higher levels of Rrp6 might artificially expand its range of targets? The authors should test whether kd of Rrp6 and other exosome components affects NCL-targets (IL1b, IL6 etc.) more than non-targets like Ccl4 and non-inflammatory mRNAs.

Line 308: It is unclear what is meant by "phase of infection". Virus-infection? Or transduction with the lentivirus for ectopic NCL expression?

Line 315: The authors claim that there is a "direct interaction" between NCL and Rrp6/exosome, although they have documented interactions only by co-IP, which may be indirect. They should either modify their statement, or demonstrate a direct interaction using recombinant proteins in vitro.

Figure 6b shows that OA treatment causes Rrp6 to localize more in the nucleolar fraction, whereas NCL is more cytoplasmic. This is not compatible with NCL being responsible for nucleolar localization of Rrp6, as the authors suggest.

The authors should not call the interaction between NCL and PP2A strong (line 323) based on the relatively weak signal in the co-IP (Figure 6c).

Line 330: The Lys296 mutant, not the "mutation of Lys296", ... "was observed in the cytoplasm..."

While sumoylated NCL is clearly visible in Figure 6d, this is not the case in Figure 6g, where an unspecific band is visible even in the negative control IP sample lacking Myc-Sumo expression.

In describing Figure 6h, the authors claim that "NCL-K296R-GFP significantly impaired decay of NCL-binding IL6, Ccl2, and Ccl5 pre-mRNAs, whereas cells overexpressing wild-type NCL-GFP successfully degraded their pre-mRNAs". However, the experiment addresses only the steady-state expression

level of the cytokine mRNAs, hence the authors should not conclude on the mRNA decay rate. In fact, there is only one experiment where mRNA decay is measured directly (Figure 3e), and it would be advisable that mRNA decay rates are assessed in some of the other essential experiments.

Extended Data Fig. 8: While OA treatment seems to confirm the authors model that NCL phosphorylation antagonizes its function in nucleolar recruitment of Rrp6/exosome, OA will affect a plethora of proteins and processes given that PP2A has a very broad target spectrum. Hence, these experiments are not specific. The authors should make more use of phospho-deficient and phospho-mimetic mutants to address the role of NCL phosphorylation in nucleolar targeting and Rrp6/exosome recruitment. The same applies to the sumoylation-deficient K296 mutant. Do these PTMs really control nucleolar localization of NCL and/or Rrp6/exosome, or only the nuclear to cytoplasmic distribution of NCL? IF analysis of the wt and mutant proteins would help to clarify this point.

Reviewer #3 (Remarks to the Author):

Lee et al present an interesting set of data that suggest that NCL and RRP6, two ribosome biogenesis factors, play a role in the metabolism of cytokine mRNAs following LPS treatment in the nucleolus. They demonstrate a condensation and fusion of the nucleoli following LPS treatment. They demonstrate that their fractionations of nucleoli are enriched in unspliced pre-mRNA encoding cytokines. Disruption of nucleolar structure by knockdown of specific proteins or drug treatment increases the levels of these mRNAs. They determine that NCL binds to the intron of these mRNAs at a C/U rich site and that knockdown of NCL increases the levels of mRNA and the half-life of pre-mRNA. They determine that NCL interacts with RRP6 and that knockdown of RRP6 and other exosome components increases the levels of knockdown increases the expression of these mRNAs. In immune cells, they demonstrate that RRP6 only localizes to the nucleolus after LPS treatment and this localization is dependent upon NCL. Finally, they demonstrate that phosphorylation and sumoylation may modulate these interactions.

In general, the manuscript is well written and each individual experiment is well controlled. The authors have demonstrated that NCL is involved, in some manner, in the regulation of cytokine mRNA metabolism. However, as a whole, revisions are needed to improve the manuscript before publication. There are several instances in which data presented appears to contradict other data in the paper. These discrepancies must be clarified either through experimentation or via better explanation (see below). More importantly, the role the relevance of NCL in is to maintaining immune homeostasis must be better clarified.

For example, it is not clear if or how having increased amounts of cytokine pre-mRNA is affecting the cell/animal or if detrimental effects are due to a loss of NCL-mediated degradation. Indeed, the closest evidence that the authors present is regarding knockdown in the liver and lung of mice, which demonstrates an increase sensitivity to LPS. However, it is not clear if this is due to increased cytokine mRNA levels or if NCL in its many other cellular roles is necessary. Clarity towards demonstrating a role and necessity for degradation is needed. For example, if degradation of these pre-mRNAs is necessary for survival during bacterial infection, then, expression of intron-less constructs of these genes from their native promoter should be detrimental to these cells after LPS treatment. This is because they could not be targeted for NCL-mediated degradation. Is that true? Is the cause of death related to increased cytokine expression? Please note that because the NCL-binding sequence may also be necessary for splicing (see below), the constructs with this deleted would not be sufficient.

The authors must better clarify which cells they are using in each experiment. Often, the authors refer to "macrophages". However, they also note that they have used RAW264.7 cells and BMDMs. When the authors refer to macrophages are they referring to isolated BMDMs or to immortal cultured RAW264.7 cells?

The authors must reword the sentence claiming that only 2% of pre-mRNA is converted into mRNA based on the publication of Brandhorst & McConkey (1974). This is not the correct interpretation of the data presented in this paper and dramatically overstates how much pre-mRNA is degraded. In the cited publication, the authors analyze nuclear hnRNA labeled by tritiated adenosine and determine that 2% of that fraction is later found on polysomes. This does not mean that 2% of pre-mRNA is processed. First, 100% of hnRNA cannot be assumed to be pre-mRNA, particularly in light of discoveries made since 1974 (e.g., the discovery of many different non-coding RNA species). Second, and perhaps most important, the authors calculation of hnRNA includes RNA that is encoded by introns, which were not discovered for another 3 years. Thus, for any pre-mRNA of ~10 kB to generate a 1 kB mRNA, 90% would be degraded. This is not because only a fraction of the synthesized pre-mRNAs are matured, but because only a small fraction of pre-mRNA is used to make a mature RNA. Granted, this is not something that the authors were aware of at the time, however, it should be evident to any reader today. Finally, Brandhorst & McConkey's data relies on mRNA that are immediately found on polysomes after synthesis. This does not account for mRNAs that are not immediately translated. The better interpretation of the data presented in Brandhorst & McConkey is that 2% of the nucleotides initially incorporated into hnRNA is later found on polysomes, presumably as mature mRNA.

I am also unclear of the relevance of the statement: "However, there is no direct evidence linking structural changes to functional diversity during infection." Hasn't this been shown, by these authors, for InSAC? If they are referring solely to the nucleolus, there is decades of data demonstrating structural changes following infection. For an older review, please see Hiscox 2007, Nature reviews Microbiology or for a more recent review, see Iarovaia et al. 2021, Cells. Structural changes to the nucleolus are also well established regarding different stimuli, particularly mitogenic signaling and stress responses. Further, the authors often state that the structural changes are affecting the activity. For example, Ln 115 – 116, the authors state, "we focused on nucleolar RNA species because structural change in this subcellular organelle has been reported to alter its RNA contents by regulating RNA synthesis and export." This is counter to most understanding of how these structures are assembled and maintained. Rather, changes in RNA synthesis and export affect the structure of these organelles. For example, treatment of cells with Actinomycin D abolishes the nucleoli concurrent with inhibition of transcription. However, inhibition of transcription causes the dissolution of the nucleoli, rather than nucleolar ablation causing the inhibition of transcription. Throughout the manuscript, the authors must be more careful in ascribing causality to events.

The authors state, "To allow for protein-coding RNA detection, highly abundant rRNAs were removed from total RNAs before global spatiotemporal RNA-sequencing (RNA-seq) analysis (Extended Data Figure 2b)". However, extended data 2b shows fractionation based on proteins localization. Where is the RNA fractionation data? If the authors do not mean to show they have depleted rRNA they should not refer to this figure. Further, in the methods, the authors only state that they conduct a ribodepletion. Thus, to state that they are analyzing protein-coding RNAs is not true, particularly with regards to the nucleolus which is concentrated with multiple RNA species. They are analyzing non-ribosomal RNAs, some of which are protein coding, others are non-coding (e.g., snoRNA, rIGS RNA, PAPAS RNA, p-RNA). Similarly, Lamin is better categorized as a nuclear membrane marker rather than a nucleoplasmic marker. Since the authors main argument concerns localization of RNA in the nucleus, the authors should provide a better nucleoplasmic marker to ensure efficient separation of nucleoplasm from the nucleolus. Data presented later in the manuscript (Figure 6b) further highlights the need to clarify the purity of these fractions. By immunofluorescence (Figure 5D), NCL is highly enriched in the nucleolus, consistent with most published data. However, western blotting of fractionated cells (figure 6B) shows equal distribution of NCL between nucleolar and nucleoplasmic

fractions. How do the authors account for these discrepancies?

The following sentence is unintelligible: "Intriguingly, approximately 48% of detectable reads in the nucleolus were found to be protein-coding RNAs which exclusively possess introns, whereas fully spliced mRNAs were strictly limited to nucleoplasm to nucleolus transport and instead were rapidly translocated to the cytoplasm for translation (Fig 2a)." It seems like a clause has been deleted at some point. Could the authors please clarify what "exclusively possess introns" means and to what the percentages refer? Do the authors suggest that half of all pre-mRNA is found in the nucleolus? If so, this highlights the need for better analysis of fractionation. There is little evidence of such pervasive transcription and splicing of mRNAs in the nucleolus, which is the site of rRNA processing and ribosome assembly. Alternatively, do the authors mean to suggest that half of the "protein-coding" RNA in the nucleolus contains introns? This is what would be suggested by the above sentence, but is not true of what is represented in the data.

With regards to figure 2a, it is also unclear what RNA is being analyzed. The figure legend states that it is a "comparison of changes in average exonic and intronic RNA levels of protein-coding genes...in CA, NP, and NL fractions at all time points of LPS." So, is this only 2, 6, 12, and 15 hrs post LPS? Or is 0 hr included? If not, is the potentially perturbed distribution of exonic/intronic RNA a result of LPS treatment?

RNA-FISH Data suggests that these RNAs are found at the periphery of the nucleolus, although, the area which was quantified obscures that fact. Are these mRNAs always found on the outer edge of the nucleolus? The authors should comment on how this pertains to their previously published data regarding "InSac". If maturation of IL-6 occurs in an "InSac" would that suggest that the InSac is a substructure within the nucleolus? In a previous publication, the InSac did not co-localize with the nucleolus, although looking at the data in that paper from the Park lab, they do appear to be peripheral to the nucleolus in the published figures. Is the InSac co-purifying with the nucleolus?

Similarly, the FISH data does not seem to correspond to the RNA-seq data. Granted, the meaning of the RNA-seq data is unclear, but if the interpretation is that half of the pre-mRNA is in the nucleolus and half is in the nucleoplasm (Fig 2A), this is not represented in the FISH data. How do the authors explain this discrepancy?

The authors suggest that it is possible that NCL is not necessary for rRNA processing in immune cells. This strains credulity without better data showing that is true. Ribosome biogenesis is an extremely conserved process not just amongst cell types, but between species. That every other cell needs NCL for rRNA maturation, but not immune cells is very tenuous. Particularly, since much of the early analysis of rRNA processing and transcription in mammalian cells were conducted using L1210 cells, which, while cancerous, were derived from immune cells (See the work of Dr. Sollner-Webb). Perhaps the discrepancy in the data arises from the fact that the KD of NCL is not as robust when analyzing rRNA (compare 2e with 2d). Given the wealth of information connecting NCL to rRNA maturation, the default position should be that NCL is required in immune cells, but the authors have not yet depleted it to a point where they affect rRNA maturation. If the authors believe that NCL is not essential for this function in immune cells, more solid data is required.

On a related note, others have shown that following LPS treatment, the transcription (and thus processing) of pre-rRNA increases >15x in B cells (Liu & Rose 1985, JBC). Thus, possibly in line with the authors supposition that NCL is not needed in immune cells, it is possible that resting cells do not require NCL, because they are not making much rRNA. This could explain the phenomena the authors see. However, this would present further complications in the interpretation of the data. Upon stimulation by LPS, ribosome biogenesis rapidly ramps up, at which point NCL becomes vital. Localization of RRP6 presented later (Figure 5) would also suggest this. Therefore, the loss of viability after LPS treatment in mouse experiments could be equally attributable to a defect in ribosome biogenesis after stimulation. Are the authors aware of any changes to the rate of rRNA synthesis (not

the steady state levels) following LPS treatment in macrophages?

Based on location and sequence composition, the C/U rich-region that the authors identify as being bound by NCL looks suspiciously like the "polypyrimidine tract" that is necessary for pre-mRNA splicing. Is that true or is this distinct from the polypyrimidine tract? If it is the polypyrimidine tract, wouldn't that suggest that NCL may be involved in the splicing of these RNAs, either positively or negatively? That is, if NCL is bound to the polypyrimidine tract, it could compete with U2AF for binding and prevent its maturation thereby altering its metabolism.

The primer set used by the authors to analyze the 47S rRNA is not specific for the 47S rRNA. The primer pair amplifies a product that is entirely 5' of the A'/01 processing site. To truly analyze the 47S rRNA, this primer set should span this processing site. Otherwise, the authors could be analyzing cleaved 1-01 fragment (i.e. the 5' cleavage product following conversion of 47S to 45S). This is most relevant later regarding RRP6, which has been shown to be necessary for degradation of 5'ETS cleavage products (Kobylecki et al 2018). Since the authors functionally link NCL with RRP6 and argue that NCL aids in RRP6-mediated degradation this is particularly relevant.

The Actinomycin D experiment presented in figure 3E has a major confounding variable, if the authors are correct in their assumption that the nucleolar structure plays a role in the stability of these RNAs. ActD treatment rapidly causes the disruption of the nucleolus. In fact, pre-rRNA transcription is inhibited prior to pre-mRNA transcription due to the G-C content of rDNA. Loss of pre-rRNA transcription leads to the dissolution of the nucleolus. This also raises another confounding variable in that others have shown that siRNA knockdown of NCL alters nucleolar structure. If the authors premise, that the nucleolus is important for the stability of these RNAs, how do they distinguish between the specific effects of NCL and the perturbation of the nucleolus. Isn't this simply the same experiment as was conducted following NPM knockdown or etoposide treatment? How can the authors distinguish between the effect of NCL knockdown or transcriptional arrest when their argument regarding nucleolar pre-mRNA regulation relies on a stable nucleolus?

The localization of Rrp6 is interesting and certainly suggests a dynamic role for this protein following LPS treatment. Previous work has shown that this protein is also necessary for rRNA processing. In fact, the name for the protein is "ribosomal RNA processing 6." Thus, it is confusing as to why Rrp6 is not localized to the nucleolus under basal conditions, as it is in other cell types. Other studies have demonstrated that this protein is localized to the nucleolus and has multiple roles in the maturation of rRNAs. In addition to NCL not being necessary for rRNA processing in immune cells, do the authors suggest that Rrp6 is also dispensable? Or is it not necessary for rRNA maturation until after the cells have been stimulated?

Could the authors clarify what is being measured in the 5th set of columns in figure 5c? Each set of columns are Il1b and Il6 pre-mRNA and mRNA. The 5th is identified as "k/d". Is this a knockdown of each individual RNA?

The data concerning the dependence of NCL:RRP6 interaction on sumoylation (figure 6F) is weak. Following LPS treatment, there is still pulldown of RRP6 with K296R-NCL, although, I agree, it does seem weaker. However, this must be quantified over multiple experiments to be convincing.

Comments from the reviewers that required a response are in bold italics, with each reply appearing in normal font just below the comment. In the replies, new material is emphasized in bold.

Reviewer #1 (Remarks to the Author):

In this manuscript, the authors report that nucleolus is an essential site of destabilization of intron-containing inflammatory RNAs during infection. Inflammatory genes have a higher read density of introns than non-inflammatory RNAs, and their pre-mRNAs are highly enriched in the nucleolus during infection. NCL was found to recruit inflammatory pre-mRNAs containing (C/U)-rich sequences and the Rrp6-exosome complex into the nucleolus by direct interaction, thereby acting as a guide factor to enable target RNA delivery to the Rrp6 exosome and subsequent degradation. In addition, depletion of NCL leads to abnormal hyperinflammation and severe lethality in response to LPS.

Furthermore, the dynamics of post-translational modification of NCL is what determines its functional activity during the LPS phase. These results represent a novel nucleolus-dependent pathway for maintaining the integrity of inflammatory gene expression and immunological homeostasis during infection.

The study is carefully constructed and clear conclusions are drawn from the individual experiments. The data can be taken as reliable. In order to improve it further, some concerns need to be addressed.

Comments

- ***On page 5, lines 110-111, Fig. 1c is included as an infection experiment, but Fig. 1c is an LPS stimulation experiment, not an infection experiment.***

We thank the reviewer for catching this error. We have amended the text in the revised manuscript.

- ***Are alveolar macrophages and Kupffer cells targeted by the lung-specific and liver-specific siRNAs shown in Figure 3c? What are the cells used in Extended Data Figure 4c?***

The in vivo jetpep and in vivo jetpei systems are broadly used for high-efficiency *in vivo* knockdown of target genes in liver- or lung-resident cells, including macrophages and dendritic cells (Jiao et al., *Nat. Immunol.*, 2015; Zhou et al., *Nat. Commun.*, 2014; Li et al., *Nat. Microbiol.*, 2016; Stellari et al., *J. Transl. Med.*, 2016; Yoshimoto et al., *Nature*, 2013; Yu et al., *Immunity*, 2016; Nakabori et al., *Sci. Rep.*, 2016; Komarov et al., *Cell Death Dis.*, 2016; Ashkenazi et al., *Nature*, 2017). To avoid any confusion, we modified the representations of each category in the graph and included detailed information in the text, Figure legend, and the Methods section of the revised manuscript.

• **It is interesting that NCL has the similar affinity for C/U-rich sequences as NCL recognition elements (NREs). Is this through the same RBD? Or can they bind in completely different forms? The NRE forms a stem-loop structure; does the C/U-rich sequence form any structure?**

The reviewer raised a valid question if the interaction of NCL with the C/U-rich motif depends on its RBD domain or if the C/U-rich sequences form any secondary structures. As previously noted, NCL Δ AS affected both nucleolar enrichment and instability of *I11b* and *I16* pre-mRNAs (**Supplementary Fig. 7c**); however, this effect was not observed in macrophages expressing a deletion mutant of NCL that lacked the RBD and RGG domains (NCL Δ RBD+RGG) (Please see the additional Figure #1 at the end of this response.). Because NCL binding of inflammatory pre-mRNAs is essential for nucleolar targeting and their degradation (**Figs. 5 and 6**), it is possible that the RBD or RGG domain of NCL is involved in the C/U-rich-containing inflammatory pre-mRNAs.

Since the C/U-rich sequences are primarily composed of cytosine and uracil residues, presumably they form a linear RNA structure. We have also tried to examine whether secondary structures exist in the C/U-rich sequences using various tools to predict RNA secondary structure; however, no secondary structures were predicted. In particular, the C/U-rich sequences are likely to contain a polypyrimidine tract, which is known in pre-mRNAs to be capable of binding various RNA-binding proteins, such as hnRNP, U2AF, or polypyrimidine tract binding protein (PTB), leading to RNA splicing (Wagner et al., *Mol. Cell Biol.*, 2001; Oberstrass et al., *Science*, 2005; Sickmier et al., *Mol. Cell*, 2006). However, the polypyrimidine tract is especially rich with uracil and is typically shorter than 20 nucleotides in length, located 5~40 base pairs before the 3' end of the intron to be spliced, whereas the C/U-rich motif contains almost even numbers of cytidines and uracils and is usually over 20 nucleotides in length, located in various intronic positions. Thus, the C/U-rich motif is presumably different from the polypyrimidine tract. Indeed, we did not observe defects of inflammatory RNA splicing in *Ncl*-knockdown cells throughout many experiments. Nevertheless, this gave us insight for further studies related to the additional roles of NCL related to cooperation or competition with U2AF or PTB in inflammatory RNA stability or splicing. We have included additional statements in the Discussion section of the revised manuscript to reflect the reviewer's points.

• **Why does Rrp6/43/46 predominantly translocate from the nucleoplasm to the nucleolus and co-localize with NCL in macrophages, even though HeLa and NIH3T3 cells also respond to LPS?**

The reviewer raised an important question by asking why Rrp6 exosome components translocate to the nucleolus only in macrophages, and not in HeLa or NIH3T3 cells. Because macrophages are major phagocytic cells with cytoskeleton dynamics that, compared with those of fibroblasts, are especially apt to enable active circular ruffling and delivery of extracellular components (Feng et al., *PNAS*, 2008), we speculated that LPS-mediated nucleolar fusion and Rrp6 nucleolar translocation may be correlated with the difference in cytoskeleton dynamics between macrophages and other cells. We therefore attempted to observe LPS-induced nucleolar fusion and Rrp6 nucleolar translocation in macrophages treated with nocodazole, an inhibitor of microtubule disruption. Intriguingly, nucleolar fusion as well as Rrp6 translocation to nucleoli were significantly attenuated by nocodazole treatment even following LPS stimulation (Please see the additional Figures at the end of this responses.) This is a good insight for future studies of the relationship between NCL-mediated inflammatory regulation and cytoskeleton dynamics; therefore, we would like to include these results only in the point-by-point response for the current manuscript. Instead, we have included additional statements in the Discussion section of the revised manuscript to reflect the reviewer's points.

• **Is there any evidence that NCL (Thr76) is a direct substrate for dephosphorylation by PP2A?**

The reviewer raised a valid point by asking if PP2A is directly involved in dephosphorylation and nucleolar translocation of NCL. We examined the effect of NCL phosphorylation in macrophages depleted of *Ppp2cb*, which is a catalytic subunit of PP2A. We observed that depletion of *Ppp2cb* resulted in a significant increase in the level of phosphorylated NCL in LPS-stimulated macrophages (New Supplementary Fig. 9a). We now believe that PP2A binds to NCL and thus promotes NCL dephosphorylation in a LPS-dependent manner, which

is essential for nucleolar targeting of NCL during LPS stimulation. This new dataset has been incorporated into **new Supplementary Fig. 9a** of the revised manuscript.

• ***Authors propose that inflammatory pre-RNAs containing C/U-rich intron sequences are transported to the nucleolus by the NCL-Rrp6 axis, but to what extent is this transport coupled to nucleoli fusion? Does the fused nucleolus then divide (original form) after the resolution of inflammation?***

The reviewer raised an important comment about nucleolar fusion dynamics during inflammatory response. We investigated changes in the pattern of nucleolar fusion in macrophages that were stimulated by LPS for 12 h and then rested for an additional 24 h in normal conditions after washout of the LPS with fresh medium (**New Supplementary Fig. 1c**). Consistent with our previous results, nucleolar fusion appeared at 12 h after LPS stimulation, with which the nucleoli increased in size in a time-dependent manner. Intriguingly, recovery from LPS-mediated inflammation allowed the cells to return to their multinucleolate form with typical nucleolar shape and size, which was similar to that in unstimulated macrophages (**New Fig. 1c**). It is possible that the capacity for nucleolar fusion may be correlated with the maximal ability for inflammatory RNA turnover. For instance, mitochondrial fusion is selectively induced by energy demand and stress, leading to maximal mitochondrial functional yield (Rossignol et al., *Cancer Res.*, 2004; Tondera et al., *EMBO J.*, 2009; Youle et al., *Science*, 2012). Similarly, infections with microbes may evoke a specific LPS response called "LPS-induced nucleolus hyper-fusion", thereby maximizing the fidelity for inflammatory cytokine RNA turnover and quality control. This is a good insight for further studies of the relationship between nucleoli fusion and NCL-Rrp6 axis-dependent inflammatory decay. We have provided the new datasets in the **new Fig. 1c** and **new Supplementary Fig. 1c** and modified the Discussion section in the revised manuscript accordingly.

• ***Check the references form (Large/small text, abbreviations).
4,8,10,12,16,31,40,44,45,47,59***

We thank the reviewer for identifying these errors. We have amended the References of the revised manuscript.

Comments from the reviewers that required a response are in bold italics, with each reply appearing in normal font just below the comment. In the replies, new material is emphasized in bold.

Reviewer #2 (Remarks to the Author):

In this manuscript, T.A. Lee et al. propose that the nucleolar localization of inflammatory cytokine pre-mRNAs serves as a mechanism to reduce cytokine expression in stimulated macrophages. This is a very intriguing hypothesis that would establish a novel level of post-transcriptional control that has not been described before. The hypothesis is based on the observation that intron-containing inflammatory cytokine pre-mRNAs are enriched in nucleoli of stimulated macrophages, the description of nucleolin (NCL) as a factor that binds to CU-rich sequences within introns of these pre-mRNA, the finding that knockdown of NCL and other nucleolar proteins causes elevated expression of inflammatory cytokine pre-mRNAs via mRNA stabilization, the discovery that NCL associates with the nuclear Rrp6-containing exosome, and a set of results indicating that this process is controlled by posttranslational modification (PTM) of NCL.

This mechanism is clearly novel and surprising since nucleoli have so far not been implicated in regulating the expression of inducible genes. In general, I am impressed by the mechanistic insight the authors were able to gain on this novel mechanism. However, I think that several of the connections need to be demonstrated with better experiments or additional controls, and I am not sure that the model put forward is fully congruent with the results presented. Several aspects need to be clarified on this intriguing hypothesis.

Major concerns:

• The data in Figure 2a-c indicate that nucleoli are enriched for intact pre-mRNAs since the read distribution seems to correspond to that of genomic DNA (more intronic reads than exonic). What is not clear to me: are these read counts normalized to the length of the underlying sequence? If so, the density of intronic reads would be higher than the density of exonic reads, which would suggest that intronic RNA alone is enriched in nucleoli (sequestration of introns after splicing?). Since it is very hard to grasp the data in Figure 2a-c, I suggest the authors depict gene tracts of some representative inflammatory and non-inflammatory genes with the read density in the three compartments. This will help to clarify which RNA species is exactly enriched in nucleoli.

The previous RNA-seq data had been normalized by the length of the underlying sequences. Nevertheless, to avoid any confusion and provide a clear conclusion, we applied quantile normalization and reanalyzed the RNA-seq datasets, which allowed us to accurately assess global changes in distributions across different samples of each fraction. Consistent with the previous results, in the nucleolus, high amounts of inflammatory pre-mRNAs were observed

after 2 or 6 h of LPS stimulation, whereas non-inflammatory pre-mRNAs were relatively low and unchanged throughout the LPS time course (**New Fig. 2b**). Moreover, inflammatory pre-mRNAs were more enriched in the nucleolus compared to the nucleoplasm; however, this difference was not observed in non-inflammatory pre-mRNAs (**New Fig. 2c**).

We agree with the reviewer's point that it is not clear whether the intronic RNAs enriched in nucleoli represent intact pre-mRNAs or just intron remnants after splicing. To address the reviewer's important question, we selected various protein-coding genes, including inflammatory *Il6*, *Il1a*, and *Ccl2* genes, and reanalyzed the RNA-seq dataset using the Integrative Genomics Viewer (IGV), which allowed us to determine the read densities of inflammatory genes in each fraction. Indeed, intact *Il6*, *Il1a*, and *Ccl2* pre-mRNAs were enriched in the nucleolus (**New Fig. 2a** and **New Supplementary Fig. 2c**). We replaced the previous RNA-seq dataset with the new datasets analyzed with quantile normalization and incorporated the IGV results into the **new Figs. 2a-c** and **new Supplementary Fig. 2c**, and revised the text to reflect the reviewer's comments.

• If pre-mRNAs are enriched in nucleoli, one would presume that they are a) protected from splicing and b) protected from degradation in this compartment. To me, this is difficult to reconcile with the proposed model that NCL causes re-localization of specific pre-mRNAs into nucleoli and targets them for Rrp6/exosome-mediated decay. Maybe the authors could use their fractionation approach to address the stability of different RNA species in the three compartments following transcriptional shut-down with Act.D. (or more elegantly, using a nascent RNA labelling approach to avoid the side effects of Act.D on nuclear organization and homeostasis). They could separately address the stability of pre-mRNA (using intron-exon boundary spanning reads (or amplicons), mRNA (using exon-exon boundary spanning reads or amplicons), and intronic RNA (using intra-intronic reads or amplicons). Obviously, the latter will also report on pre-mRNA. This could help to clarify whether pre-mRNAs are rapidly degraded or stable in nucleoli, and also give information on the decay of intronic RNA after splicing (which is thought to be degraded rapidly in the nucleoplasm). To me, the nature and stability of the observed RNA-pol II transcripts in nucleoli are the main aspects the authors should clarify before postulating a specific model.

The reviewer raised a valuable point by asking if inflammatory pre-mRNAs are indeed degraded (or stable) in the nucleolus. As the reviewer suggested, we performed the Act.D experiment again with the nucleolar or nucleoplasmic fractions and also used specific primer pairs for intron-exon and intra-intronic regions. We clearly observed that the turnover rate of *Il1b*, *Il6*, *Cxcl2*, *Ccl2*, *Ccl4*, and *Ccl5* pre-mRNAs was much faster in the nucleolar fraction than in the nucleoplasmic fraction (**New Fig. 3f** and **New Supplementary Fig. 3f**). On the basis of these results together with the IGV results as noted above, we are now confident that inflammatory pre-mRNAs are translocated to the nucleolus and degraded via an NCL-Rrp6

axis-dependent mechanism, which is dependent on the introns of the target genes. We have incorporated the updated datasets in the **new Fig. 3f** and **new Supplementary Fig. 3f** of the revised manuscript.

• If the authors chose to measure nucleolus-associated RNA decay by RNA-Seq, they could also interrogate their data for evidence of 3'-to-5' degradation (i.e. an accumulation of reads towards the 5' end), which would be a good indication for exosome-mediated decay.

The reviewer raised an interesting point about the possibility of 3'-to-5' degradation evidenced by an accumulation of reads towards the 5' end of RNAs in each fraction. Although we found no accumulation pattern of 5' intronic reads based on our IGV analysis, the reviewer's comment gave us insight for further studies related to the mechanism underlying the involvement of other nucleolar RNA-binding proteins or nucleases, such as XRN1, in nucleolus-associated RNA degradation.

• Another important point is what the FISH signal of inflammatory pre-mRNAs in (or at the rim of) nucleoli means. Most likely, the observed foci correspond to the sites of transcription - this should be stated in the text. Does this mean that RNA-pol II is active in (or at the edge of) nucleoli? This would be very intriguing. Or is the relocalization of the sites of transcription of inflammatory genes into nucleoli a means of suppressing transcription, given that nucleoli are thought to be sites of privileged RNA-pol I activity (this could fit the idea that inflammatory genes need to be turned off at later stages of macrophage activation). For this question, it would also be important to show where the sites of transcription of control non-inflammatory and non-induced genes are. This aspect might be an important part of the authors' model, and should also be discussed in view of what is known about RNA-pol II in nucleoli.

The reviewer raised an important question about the specific sites of inflammatory or non-inflammatory pre-mRNAs associated with RNA-pol II-mediated transcription activity under physiological conditions or during later stages of infection. According to previous reports (Quinodoz et al., *Cell*, 2018; Padeken et al., *Curr. Opin. Cell Biol.*, 2014; Morf et al., *Nat. Biotech.*, 2019), actively transcribed regions of DNA are positioned away from the nucleolus; in other words, specific sites that are closer to the nucleolus tend to be transcriptionally inactive. To thus examine the spatial preferences of non-inflammatory and inflammatory pre-mRNA foci in response to LPS stimulation, we performed pre-mRNA FISH analysis with probes specific for introns of inflammatory or non-inflammatory genes and then combined the FISH signals with immunofluorescence for nucleoli (FBL-GFP) and active RNA-pol II (anti-pS2; Ser-2 phosphorylation of RNA-pol II, a marker of active RNA-pol II).

Intriguingly, *Il6* and *Il1b* pre-mRNA foci (inflammatory genes) overlapped nucleoli but were positioned away from active RNA-pol II regions in 12 h LPS-stimulated cells, while *Hmgal* pre-mRNA foci (a non-inflammatory gene) were preferentially positioned in the active RNA-pol II regions and excluded from the nucleolar compartment (**White arrows in new Fig. 3d, i and iii**). We also observed inflammatory pre-mRNA foci-pS2 contacts in the nucleoplasm,

they did not appear in nucleolar regions (**Yellow arrows in new Fig. 3d, ii and iv**). Importantly, this preferential organization of pre-mRNA foci of inflammatory genes, but not non-inflammatory genes, occurred in a LPS-dependent manner (**New Fig. 3d**).

In particular, TDP-43-mediated InSAC (for inflammatory *Il6* splicing region) was located close to, but not overlapping, the nucleolus, unlike the inflammatory pre-mRNA foci that contacted or overlapped the nucleolus. Consistent with the confocal results, purified nucleoli did not contain TDP-43 (**New Supplementary Figs. 3a and 3b**). Thus, these results indicate that InSAC regions are positioned separately from the nucleolar region. Together with the results mentioned above, we believe that the nucleolus is a site for inflammatory pre-mRNA decay rather than transcription or splicing. This is a good insight for further studies of the relationship between transcription, nucleoli, and inflammatory RNA foci, or between inflammatory RNA foci and InSAC during infection. Therefore, we have incorporated **new Fig. 3d** and **new Supplementary Figs. 3a-b**, and included additional statements in the revised manuscript to reflect the reviewer's points.

• To me, the discussion is superficial. The authors should address which RNA species is targeted to (or synthesized in) nucleoli, and what the physiological meaning of the newly described NCL-Rrp6/exosome decay pathway is. Do their data fit to earlier reports of NCL acting on mRNA stability? Is what they discovered unique to stimulated macrophages, and could it be involved in the attenuation of inflammatory responses, an important function macrophages fulfill after triggering inflammation?

We agree with the reviewer's comments and have extensively modified the Discussion section to reflect all the reviewer's points and suggestions.

Specific comments:

• Intro line 63: "...the fact only 2% of nuclear pre-mRNAs are converted to mRNAs suggests that incomplete or remnant pre-mRNAs must be actively degraded in the nucleus." This statement is misleading. The study by Brandhorst & McConkey (J Mol Biol, 1974) estimated the turnover rate of pre-mRNA based on radioactive nucleotide incorporation, and essentially reflects the fact that intron are rapidly degraded in the nucleus, while only a small portion of the pre-mRNA sequence (the exons in mRNA) is exported to the cytoplasm and loaded on polysomes. These measurements were global and did not address incomplete or remnant pre-mRNAs.

We thank the reviewer for catching inappropriate terms. We have modified the text throughout the revised manuscript per the reviewer's points.

• Figure 1 shows the growth in size and reduced numbers of nucleoli upon stimulation of macrophages: since macrophages become more flat upon activation, the change in cell shape alone could explain the increase in nucleolar area. Hence, the authors should

measure the volume of nucleoli.

We agree with the reviewer's comments. We quantified the volume of nucleoli in the confocal images using the Ellipsoid formula in the ZEN core software and incorporated graphs representing all quantifications (**New Fig. 1b** and **New Supplementary Fig. 1a**).

• *Fibrillarin is a marker for the dense fibrillar component (DFC) of nucleoli. What happens to the other components: the fibrillar center (FC) and the granular component (GC)?*

Per the reviewer's suggestion, we examined the morphological alteration of nucleoli in response to LPS stimulation by observing markers of the FC and the GC with anti-UBF and anti-NCL antibodies, respectively. Similar to the nucleolar patterns observed with fibrillarin-GFP, LPS-mediated nucleolar fusion was also observed with the FC and GC markers (**New Supplementary Fig. 1b**).

• *In Figure 2b, the intron/exon ratio in NP of non-inflammatory and inflammatory genes looks similar, whereas Figure 2d shows that this ratio is drastically different between non-inflammatory and inflammatory genes. The data seem to be contradictory. What is also important for this comparison: were the read numbers normalized to the length of the underlying sequence? If not, could it be that the length distribution of introns and exons is different between non-inflammatory and inflammatory genes?*

We agree with the reviewer's point about the discrepancy. As previously mentioned, because our previous RNA-seq analysis of non-inflammatory and inflammatory gene expression did not match well with the comparison of gene expression levels across different samples of each fraction, our approach might have been biased and led to contradictory findings. To clarify this question, we applied quantile normalization and reanalyzed the RNA-seq datasets. Consistently, inflammatory pre-mRNAs were more enriched in the nucleolus than in the nucleoplasm, whereas there was no difference in non-inflammatory pre-mRNAs between the compartments. Therefore, we replaced the previous RNA-seq dataset with the new datasets in **new Figs. 2b** and **2c** of the revised manuscript and revised the text to reflect the reviewer's comments.

• *Line 137: "Furthermore, changes in nucleolar inflammatory pre-mRNA levels did mirror that in nucleoplasmic pre-mRNA levels without further reduction of the corresponding cytoplasmic mRNA levels, indicating that these changes might not be due to attenuation of RNA splicing (Extended Data Fig. 2c)." I simply don't understand this sentence, it is way too long and complicated. To me, Extended Data Fig. 2c shows that inflammatory genes are induced in all three compartments, though earlier in NP and NL than in CA. The authors should make an effort to express their ideas in simple terms.*

We modified the text throughout the revised manuscript.

• *Line 124, Figure 2a: "approximately 48% of detectable reads in the nucleolus were found*

to be protein-coding RNAs which exclusively possess introns..." This is an overstatement, the NL fraction still contains about 10% reads from exons. The authors should phrase their observations more carefully, here e.g. that intronic RNA was strongly enriched in the nucleolar fraction. Since exons make up less than 10% of pre-mRNA sequences, this observation is compatible with stable pre-mRNAs being present in NL. This point could be further strengthened by showing the read distribution on gene tracks of a few exemplary inflammatory and non-inflammatory genes (see comment above).

We agree with the reviewer's comments. As previously mentioned, we have provided the new dataset in **new Figs. 2a-c** and **new Supplementary Fig. 2c** of the revised manuscript, and we have modified the text through the revised manuscript to reflect the reviewer's points and suggestions.

• *Figure 2e: It seems that the FISH signal of the inflammatory pre-mRNAs is at the edge of nucleoli. Is this seen in all the images? The localization should be quantified over a large number of images. Also, it would be important to show the FISH signal of a few non-inflammatory genes (see also comment above).*

We agree with the reviewer's comments. We quantified the FISH signals of the inflammatory and non-inflammatory pre-mRNAs across a large number of images and provided new graphs in **new Fig. 3b**. As mentioned above, we also combined the additional FISH datasets with dynamics of inflammatory and non-inflammatory pre-mRNA foci associated with RNA-pol II activity in response to LPS, as shown in the **new Fig. 3d** of the revised manuscript.

• *Figure 3c: In which tissue was RNA expression measured?*

We used liver and lung tissues to measure RNA expression of each gene. To avoid confusion, we added more detailed information in the Figure legends of the revised manuscript.

• *What happens to the inflammatory pre-mRNA foci (i.e. the sites of transcription) after kd of Npm1 or Ncl? This could be tested using e.g. rRNA as a marker of nucleoli, fibrillarin or a component of the exosome.*

The reviewer raised an intriguing question about alterations of the size, morphology, or spatial preference of the inflammatory pre-mRNA foci in *Npm1*-depleted cells. Similar to control cells, inflammatory *Il6* pre-mRNA foci were positioned near FBL-GFP signals, away from active RNA-pol II in *Npm1*-depleted cells. Intriguingly, both the size and fluorescence intensity of inflammatory *Il6* foci were significantly increased in *Npm1*-depleted cells compared to control cells (**New Supplementary Fig. 3d-e**). This indicates the nucleolus is involved in regulating inflammatory pre-mRNA stability rather than transcription. To support these results, we separated nucleolar and nucleoplasmic fractions from 12 h LPS-stimulated RAW 264.7 cells and examined the half-life of inflammatory pre-mRNAs by using primers for both intron-exon and intra-intronic regions after treatment with the transcription inhibitor, actinomycin D (Act.D). Notably, the turnover rate of *Il1b*, *Il6*, *Cxcl2*, *Ccl2*, *Ccl4*, and *Ccl5*

pre-mRNAs was much faster in the nucleolar fraction than in the nucleoplasmic fraction (**New Fig. 3f** and **New Supplementary Fig. 3f**). Together with the previous responses, these results support that the nucleolus, or NCL, is involved in the regulation of inflammatory pre-mRNA stability rather than the attenuation of transcription. We have incorporated **new Fig. 3f** and **new Supplementary Figs. 3d-f** and included additional statements in the revised manuscript to reflect the reviewer's points.

• **Figure 4f: How was binding of the reporter mRNA to NCL measured? Is this an RNA-IP approach, or does it include crosslinking? More detail should be given in the text.**

We thank the reviewer for catching this lack of information. We performed RNA-IP-qPCR (RIP-qPCR) experiments to measure the NCL binding affinity for the *Myc*-tagged minigenes. We have incorporated detailed information about the RNA-IP procedure in the Figure legends and the Methods section of the revised manuscript.

• **Similar to the *Myc*-minigene mRNA, the authors should test by qPCR if endogenous inflammatory pre-mRNAs are bound to NCL, using a number of non-inflammatory pre-mRNAs as controls. This would help to validate the CLIP data in Figure 4a.**

We agree with reviewer's comment. To confirm the PAR-CLIP results, we performed RIP-qPCR experiments to detect the interaction of NCL with endogenous inflammatory pre-mRNAs. The results confirmed the interaction of NCL with target inflammatory pre-mRNAs, but not with non-inflammatory pre-mRNAs. We included the new dataset in the **new Supplementary Fig. 6b** of the revised manuscript.

• **Line 256: Subtitle should say: NCL delivers inflammatory RNAs to the Rrp6-exosome for THEIR degradation**

We thank the reviewer for catching this error. We have incorporated the subtitle in the revised manuscript.

• **Extended Data Fig. 6d shows that NCL-deltaAS is less active than full-length NCL, especially towards suppressing the expression of IL6 mRNA. Hence, I do not agree with the authors' assertion that "we alternatively used the deletion mutant of NCL lacking acidic stretches (NCLdeltaAS-GFP) as its effects on both nucleolar enrichment and instability of Il1b and Il6 pre-mRNAs were similar to those of full-length NCL in control as well as Ncl-depleted macrophages (line 267).**

We agree with reviewer's comment. We have carefully amended the text to appropriately interpret the result in the revised manuscript.

• **Figure 5a/b: The finding that NCL appears to interact with Rrp6 only in LPS-stimulated cells is very interesting. However, the authors should also show the control IP (anti-GFP) in LPS-stimulated cells.**

We agree with reviewer's comment. We have added a depiction of the original silver staining image with the control IP in the revised manuscript (New Fig. 6a).

• **As a consequence of this finding, one would predict that kd of NCL causes upregulation of inflammatory mRNAs only in stimulated macrophages, but not under basal conditions. This should be tested systematically for a range of inflammatory mRNAs. Vice-versa, overexpression of NCL should suppress inflammatory mRNAs only in stimulated, but not unstimulated macrophages.**

We agree with the reviewer's suggestion to show the inflammatory RNA levels under basal conditions. We therefore conducted an RT-qPCR analysis of unstimulated macrophages that stably expressed NCL-GFP or an shRNA against *Ncl*. As expected, the unstimulated macrophages barely expressed inflammatory RNAs regardless of NCL expression levels (New Fig. 4a). We incorporated this result in new Fig. 4a of the revised manuscript.

• **Since NCL kd affects *IL1b* and *IL6*, but not *Ccl4* mRNA expression (Figure 3c), whereas *Rrp6* overexpression reduces the expression of all three mRNAs (Extended Data Figure 7b), I am concerned about the specificity of the NCL-exosome connection. Maybe *Rrp6* overexpression is not a good approach since higher levels of *Rrp6* might artificially expand its range of targets? The authors should test whether kd of *Rrp6* and other exosome components affects NCL-targets (*IL1b*, *IL6* etc.) more than non-targets like *Ccl4* and non-inflammatory mRNAs.**

The reviewer's comment highlights the important issue of clarifying the role of the NCL-exosome axis in the degradation of specific RNA targets. We have previously shown *Il1b* and *Il6* RNA expression in *Rrp6*-depleted, *Rrp43*-depleted, or *Rrp46*-depleted macrophages in Fig. 5c and Supplementary Fig. 7b of the original manuscript (Please also see the Figs. #1 and #2 at the end of this response.). Importantly, depletion of each exosome component resulted in a significant increase in *Il1b* and *Il6* RNA expression that was reversible by *Rrp6* overexpression. These figures are positioned in Fig. 6c and Supplementary Fig. 8b of the revised manuscript.

- **Line 308: It is unclear what is meant by "phase of infection". Virus-infection? Or transduction with the lentivirus for ectopic NCL expression?**

We agree with the reviewer's comment. To avoid confusion, we reworded this term in the revised manuscript.

- **Line 315: The authors claim that there is a "direct interaction" between NCL and Rrp6/exosome, although they have documented interactions only by co-IP, which may be indirect. They should either modify their statement, or demonstrate a direct interaction using recombinant proteins in vitro.**

We agree with the reviewer and thus replaced "direct interaction" with "physical interaction" throughout the revised manuscript.

- **Figure 6b shows that OA treatment causes Rrp6 to localize more in the nucleolar fraction, whereas NCL is more cytoplasmic. This is not compatible with NCL being responsible for nucleolar localization of Rrp6, as the authors suggest.**

There seems to have been a misleading. As shown in Fig. 6b of the original manuscript, Rrp6 failed to translocate to the nucleolus in OA-treated cells even with LPS stimulation, which was compatible with the result showing the failure of NCL nucleolar translocation (Please see the red or blue box representing the degree of Rrp6 or NCL nucleolar translocation, respectively, in the graphs of Fig. 6b attached at the end of this response). To avoid any confusion, we have modified the text and the proposed model in the revised manuscript to make them clearer.

- **The authors should not call the interaction between NCL and PP2A strong (line 323) based on the relatively weak signal in the co-IP (Figure 6c).**

We agree with the reviewer's point and have modified the term in the revised manuscript.

- **Line 330: The Lys296 mutant, not the "mutation of Lys296", ... "was observed in the cytoplasm..."**

We agree with the reviewer's point and have corrected the text of the revised manuscript.

• ***While sumoylated NCL is clearly visible in Figure 6d, this is not the case in Figure 6g, where an unspecific band is visible even in the negative control IP sample lacking Myc-Sumo expression.***

We agree the reviewer's point. We repeated the experiment several times and now provide more convincing and sharper images in the **new Fig. 7g** of the revised manuscript.

• ***In describing Figure 6h, the authors claim that "NCL-K296R-GFP significantly impaired decay of NCL-binding Il6, Ccl2, and Ccl5 pre-mRNAs, whereas cells overexpressing wild-type NCL-GFP successfully degraded their pre-mRNAs". However, the experiment addresses only the steady-state expression level of the cytokine mRNAs, hence the authors should not conclude on the mRNA decay rate. In fact, there is only one experiment where mRNA decay is measured directly (Figure 3e), and it would be advisable that mRNA decay rates are assessed in some of the other essential experiments.***

We agree with the reviewer's comment. We have carefully amended and modified the text of the revised manuscript to reflect the reviewer's points.

• ***Extended Data Fig. 8: While OA treatment seems to confirm the authors model that NCL phosphorylation antagonizes its function in nucleolar recruitment of Rrp6/exosome, OA will affect a plethora of proteins and processes given that PP2A has a very broad target spectrum. Hence, these experiments are not specific. The authors should make more use of phospho-deficient and phospho-mimetic mutants to address the role of NCL phosphorylation in nucleolar targeting and Rrp6/exosome recruitment. The same applies to the sumoylation-deficient K296 mutant. Do these PTMs really control nucleolar localization of NCL and/or Rrp6/exosome, or only the nuclear to cytoplasmic distribution of NCL? IF analysis of the wt and mutant proteins would help to clarify this point.***

The reviewer has raised a valid point by asking if PP2A is directly involved in dephosphorylation and nucleolar translocation of NCL. We examined the effect of NCL phosphorylation in macrophages depleted of *Ppp2cb*, which is a catalytic subunit of PP2A. Depletion of *Ppp2cb* resulted in a significant increase in the level of phosphorylated NCL in LPS-stimulated macrophages (**New Supplementary Fig. 9a**). On the basis of this data with the previous results, we now believe that PP2A binds to NCL, leading to LPS-dependent NCL dephosphorylation, which is essential for nucleolar targeting of NCL during LPS stimulation.

To further determine whether NCL sumoylation is important for NCL nucleolar localization, we conducted an IF analysis of NCL-K296R-GFP-expressing cells. Unlike wild-type NCL, NCL-K296R-GFP was considerably detected in the cytoplasm even in the presence of LPS, which is consistent with a previous report (**New Supplementary Fig. 9f**; Zhang et al., *J Biol*

Chem., 2015). Together with the immunoblots of the nuclear fraction in the **new Supplementary Fig. 9e**, these results suggest that NCL sumoylation is critical for NCL nuclear translocation, but not for NCL nucleolar translocation. Therefore, these new datasets have been incorporated into the **new Supplementary Figs. 9a** and **9f** of the revised manuscript.

Comments from the reviewers that required a response are in bold italics, with each reply appearing in normal font just below the comment. In the replies, new material is emphasized in bold.

Reviewer #3 (Remarks to the Author):

Lee et al present an interesting set of data that suggest that NCL and RRP6, two ribosome biogenesis factors, play a role in the metabolism of cytokine mRNAs following LPS treatment in the nucleolus. They demonstrate a condensation and fusion of the nucleoli following LPS treatment. They demonstrate that their fractionations of nucleoli are enriched in unspliced pre-mRNA encoding cytokines. Disruption of nucleolar structure by knockdown of specific proteins or drug treatment increases the levels of these mRNAs. They determine that NCL binds to the intron of these mRNAs at a C/U rich site and that knockdown of NCL increases the levels of mRNA and the half-life of pre-mRNA. They determine that NCL interacts with RRP6 and that knockdown of RRP6 and other exosome components increases the levels of knockdown increases the expression of these mRNAs. In immune cells, they demonstrate that RRP6 only localizes to the nucleolus after LPS treatment and this localization is dependent upon NCL. Finally, they demonstrate that phosphorylation and sumoylation may modulate these interactions.

In general, the manuscript is well written and each individual experiment is well controlled. The authors have demonstrated that NCL is involved, in some manner, in the regulation of cytokine mRNA metabolism. However, as a whole, revisions are needed to improve the manuscript before publication. There are several instances in which data presented appears to contradict other data in the paper. These discrepancies must be clarified either through experimentation or via better explanation (see below). More importantly, the role the relevance of NCL in is to maintaining immune homeostasis must be better clarified. For example, it is not clear if or how having increased amounts of cytokine pre-mRNA is affecting the cell/animal or if detrimental effects are due to a loss of NCL-mediated degradation. Indeed, the closest evidence that the authors present is regarding knockdown in the liver and lung of mice, which demonstrates an increase sensitivity to LPS. However, it is not clear if this is due to increased cytokine mRNA levels or if NCL in its many other cellular roles is necessary. Clarity towards demonstrating a role and necessity for degradation is needed. For example, if degradation of these pre-mRNAs is necessary for survival during bacterial infection, then, expression of intron-less constructs of these genes from their native promoter should be detrimental to these cells after LPS treatment. This is because they could not be targeted for NCL-mediated degradation. Is that true? Is the cause of death related to increased cytokine expression?

Please note that because the NCL-binding sequence may also be necessary for splicing (see below), the constructs with this deleted would not be sufficient.

We thank the reviewer for giving us valuable and important comments. We have carefully addressed all the reviewer's questions and issues on a point-by-point basis.

The reviewer raised a concern that increased inflammatory cytokine RNA levels indeed accelerate detrimental effects on cells or animals. It is well known that failure of inflammatory cytokine RNA decay is closely associated with severe autoimmune diseases, including septic shock/death (Please see the Figs. #1 and #2 attached to the end of this response that are derived from parts of many references; Matsushita et al., *Nature*, 2009; Uehata et al., *Cell*, 2013; Pratama et al., *Immunity*, 2013; Nakatsuka et al., *Mucosal Immunol.*, 2018). Importantly, because inhibition of pre-mRNA degradation leads to the accumulation of spliced mRNAs and, eventually, increased mRNA production, many researchers have tried to understand the mechanisms that regulate RNA metabolism and homeostasis (Lemieux et al., *Mol. Cell*, 2011; Bousquet-Antonelli et al., *Cell*, 2000; Bergeron et al., *Mol. Cell Biol.*, 2015). We are therefore excited to provide evidence for a new pathway of NCL-Rrp6 axis-dependent degradation of inflammatory pre-mRNAs, which occurs specifically in the nucleolus.

[redacted]

[redacted]

The reviewer also raised concerns about whether decreased survival due to *Ncl* knockdown is related to increased cytokine mRNA levels or to one of the many other cellular roles of NCL. Because NCL is a multi-functional protein that regulates rRNA biogenesis, transcription, splicing, and cell division/apoptosis, the reviewer suggested additional experiments using intron-less constructs to clarify this issue. It has been difficult for us to decide which or how many intronic regions to delete, however, because introns may be necessary for splicing, stability, or decay of the target RNA. Therefore, we addressed this issue with several datasets demonstrating that:

1. There were no detrimental effects of *Ncl* knockdown in mice or cells. Because a complete knockout of *Ncl* is lethal, we have tried to partially deplete *Ncl* using shRNA or siRNA in cells or mice, respectively. *Ncl* expression was successfully reduced in the *Ncl*-knockdown mice and cells, but 30~50% of total cells still expressed *Ncl*, as shown in **additional Figs. #1 and #2** at the end of this response. We observed no considerable harmful effects of *Ncl* knockdown on cell viability or mouse survival, which suggests that the remaining *Ncl* expression was enough to enable NCL functions such as apoptosis, rRNA biogenesis, transcription, and splicing (Additional evidence for each function is shown below).
2. There were no significant effects of *Ncl* knockdown on apoptosis. We examined the effect of *Ncl* knockdown on the cell cycle and apoptosis and found that neither cell-cycle arrest nor apoptosis was detectable in *Ncl*-knockdown macrophages. (Please see the **additional Fig. #3**.)
3. There were no significant effects of *Ncl* knockdown on rRNA biogenesis. Because NCL is an essential factor for rRNA biogenesis, we performed RT-qPCR analysis to determine if 28S and 47S rRNA expression levels were impaired by *Ncl* knockdown in our system. To provide more convincing results, we used different titers of retrovirus expressing *Ncl* shRNA, which allowed us to assess the effect of the degree of *Ncl* knockdown on rRNA expression. Consistently, we found that neither 28S nor 47S rRNA levels were changed in *Ncl*-knockdown cells regardless of the degree of knockdown, whereas inflammatory *Il6* pre-mRNA levels were significantly increased in the *Ncl*-knockdown cells. (Please see the **additional Fig. #4**.) Nevertheless, we agree with the reviewer's concerns, and we revised the text to reflect the reviewer's point.
4. There were no significant effects of *Ncl* knockdown on RNA splicing. Because NCL is involved in the RNA splicing process, we revisited RT-PCR results for various inflammatory genes in *Ncl*-knockdown cells. We did not observe any effects of *Ncl* knockdown on the expression of the *Il1b*, *Il6*, *Cxcl2*, or *Ccl2* genes. Moreover, if the *Ncl* knockdown affected RNA splicing, most mRNA levels would generally decrease with *Ncl* knockdown; however, we consistently observed increased mRNA levels in the *Ncl*-knockdown cells. (Please see the **additional Fig. #5**.)
5. There were no significant effects of *Ncl* knockdown on promoter activity/transcription. This was not due to any effect on transcriptional efficiency, because no differences were

seen in the promoter activity of *Il6* in *Ncl*-knockdown cells. (Please see the **additional Fig. #6.**)

6. We also attempted to verify the specific location of inflammatory pre-mRNA foci associated with transcriptional activity during LPS stimulation. Previous reports (Quinodoz et al., *Cell*, 2018; Padeken et al., *Curr. Opin. Cell Biol.*, 2014; Morf et al., *Nat. Biotech.*, 2019) showed that actively transcribed regions are positioned away from the nucleolus; in other words, sites that are closer to the nucleolus tend to be transcriptionally inactive. To thus examine the spatial preferences of non-inflammatory and inflammatory pre-mRNA foci in response to LPS stimulation, we performed pre-mRNA FISH analysis with probes specific for introns of inflammatory or non-inflammatory genes and then combined the FISH signals with immunofluorescence for nucleoli (FBL-GFP) and active RNA-pol II (anti-pS2; Ser-2 phosphorylation of RNA-pol II, a marker of active RNA-pol II). Intriguingly, *Il6* and *Il1b* pre-mRNA foci (inflammatory genes) overlapped nucleoli but were positioned away from active RNA-pol II regions in 12 h LPS-stimulated cells, while *Hmgal* pre-mRNA foci (a non-inflammatory gene) were preferentially positioned in the active RNA-pol II regions and excluded from the nucleolar compartment (**White arrows in new Fig. 3d, i and iii**). We also observed inflammatory pre-mRNA foci-pS2 contacts in the nucleoplasm, they did not appear in nucleolar regions (**Yellow arrows in new Fig. 3d, ii and iv**). Importantly, this preferential organization of pre-mRNA foci of inflammatory genes, but not non-inflammatory genes, occurred in a LPS-dependent manner (**New Fig. 3d**).

In particular, TDP-43-mediated InSAC (for inflammatory *Il6* splicing region) was located close to, but not overlapping, the nucleolus, unlike the inflammatory pre-mRNA

foci that contacted or overlapped the nucleolus. Consistent with the confocal results, purified nucleoli did not contain TDP-43 (**New Supplementary Figs. 3a and 3b**). Thus, these results indicate that InSAC regions are positioned separately from the nucleolar region. Together with the results mentioned above, we believe that the nucleolus is a site for inflammatory pre-mRNA decay rather than transcription or splicing.

We hope that the updated results address the reviewer's concerns and show that the nucleolus acts as a site for NCL-Rrp6 axis-dependent inflammatory pre-mRNA decay. Nevertheless, we agree with the reviewer's concerns and have incorporated the important insights (for further studies of the relationship between transcription, nucleoli, and inflammatory RNA foci, or between inflammatory RNA foci and InSAC during infection) and possible limitations of our studies raised by the reviewer in the text of the revised manuscript.

• *The authors must better clarify which cells they are using in each experiment. Often, the authors refer to “macrophages”. However, they also note that they have used RAW264.7 cells and BMDMs. When the authors refer to macrophages are they referring to isolated BMDMs or to immortal cultured RAW264.7 cells?*

We agree with the reviewer's comment about the ambiguous term for the cells used in each experiment. In the revised manuscript, we give the specific names of the cells in the text, all Figures and Figure legends.

• *The authors must reword the sentence claiming that only 2% of pre-mRNA is converted into mRNA based on the publication of Brandhorst & McConkey (1974). This is not the correct interpretation of the data presented in this paper and dramatically overstates how much pre-mRNA is degraded. In the cited publication, the authors analyze nuclear hnRNA labeled by tritiated adenosine and determine that 2% of that fraction is later found on polysomes. This does not mean that 2% of pre-mRNA is processed. First, 100% of hnRNA cannot be assumed to be pre-mRNA, particularly in light of discoveries made since 1974 (e.g., the discovery of many different non-coding RNA species). Second, and perhaps most important, the authors calculation of hnRNA includes RNA that is encoded by introns, which were not discovered for another 3 years. Thus, for any pre-mRNA of ~10 kB to generate a 1 kB mRNA, 90% would be degraded. This is not because only a fraction of the synthesized pre-mRNAs are matured, but because only a small fraction of pre-mRNA is used to make a mature RNA. Granted, this is not something that the authors were aware of at the time, however, it should be evident to any reader today. Finally, Brandhorst & McConkey's data relies on mRNA that are immediately found on polysomes after synthesis. This does not account for mRNAs that are not immediately translated. The better interpretation of the data presented in Brandhorst & McConkey is that 2% of the nucleotides initially incorporated into hnRNA is later found on polysomes, presumably as mature mRNA.*

We thank the reviewer for catching our inappropriate statements and giving kind and detailed explanations. We agree with the reviewer's comment and have carefully amended the text throughout the revised manuscript.

• I am also unclear of the relevance of the statement: “However, there is no direct evidence linking structural changes to functional diversity during infection.” Hasn’t this been shown, by these authors, for InSAC? If they are referring solely to the nucleolus, there is decades of data demonstrating structural changes following infection. For an older review, please see Hiscox 2007, Nature reviews Microbiology or for a more recent review, see Iarovaia et al. 2021, Cells. Structural changes to the nucleolus are also well established regarding different stimuli, particularly mitogenic signaling and stress responses. Further, the authors often state that the structural changes are affecting the activity. For example, Ln 115 – 116, the authors state, “we focused on nucleolar RNA species because structural change in this subcellular organelle has been reported to alter its RNA contents by regulating RNA synthesis and export.” This is counter to most understanding of how these structures are assembled and maintained. Rather, changes in RNA synthesis and export affect the structure of these organelles. For example, treatment of cells with Actinomycin D abolishes the nucleoli concurrent with inhibition of transcription. However, inhibition of transcription causes the dissolution of the nucleoli, rather than nucleolar ablation causing the inhibition of transcription. Throughout the manuscript, the authors must be more careful in ascribing causality to events.

We thank the reviewer for catching these inappropriate statements. We have carefully amended the text throughout the revised manuscript.

• The authors state, “To allow for protein-coding RNA detection, highly abundant rRNAs were removed from total RNAs before global spatiotemporal RNA-sequencing (RNA-seq) analysis (Extended Data Figure 2b)”. However, extended data 2b shows fractionation based on proteins localization. Where is the RNA fractionation data? If the authors do not mean to show they have depleted rRNA they should not refer to this figure.

We thank the reviewer for catching this mislabeling and lack of information. We have carefully amended these errors throughout the revised manuscript.

Further, in the methods, the authors only state that they conduct a ribodepletion. Thus, to state that they are analyzing protein-coding RNAs is not true, particularly with regards to the nucleolus which is concentrated with multiple RNA species. They are analyzing non-ribosomal RNAs, some of which are protein coding, others are non-coding (e.g., snoRNA, rIGS RNA, PAPAS RNA, p-RNA).

We would have liked to investigate whether protein-coding transcripts are present in the nucleolus or if their subcellular abundance and distribution are influenced in a time course of LPS stimulation. As the reviewer has pointed out, many different types of RNAs are present in the nucleolus. However, because rRNA is the most highly abundant component of total

RNA isolated from cells, comprising the majority (> 80~90%) of the molecules in a total RNA sample (O'Neil et al., *Curr. Protoc. Mol. Biol.*, 2013), we needed to remove rRNAs from total RNA before RNA-seq analysis was performed. This allowed us to enable efficient protein-coding transcript detection, thereby assessing global changes in distributions or expression levels of non-inflammatory or inflammatory pre-mRNAs across different samples. To avoid any confusion, we have properly reworded the text throughout the revised manuscript.

Similarly, Lamin is better categorized as a nuclear membrane marker rather than a nucleoplasmic marker. Since the authors main argument concerns localization of RNA in the nucleus, the authors should provide a better nucleoplasmic marker to ensure efficient separation of nucleoplasm from the nucleolus.

The reviewer raised a concern about the use of Lamin as a nucleoplasmic marker. Several previous reports used immunoblot with anti-Lamin to verify successful isolation of the nucleoplasmic fraction from nucleoli (Please see the Fig. #1 attached in the end of this response that are derived from references following; Andersen et al., *Curr. Biol.*, 2002; Boisvert et al., *Mol. Cell Proteomics*, 2010) Therefore, we followed that method to ensure efficient separation. When nucleoli are isolated from nuclei, because the nuclei are physically disrupted by sonication, the nucleoplasmic fraction generally contains some nuclear integral or peripheral proteins, such as Lamin. Nevertheless, we agree with the reviewer's concern and therefore performed a reciprocal immunoblot analysis with H2AX, a typical nucleoplasmic marker that is absent at the nuclear envelope. Consistently, H2AX was observed only in the nucleoplasmic fraction and not in the cytoplasmic and nucleolar fractions (Please see the **additional Fig. #2** at the end of this response). To further confirm this result, we examined the amount of 18S rRNA, which is rapidly transported to the nucleoplasm from the nucleolus after processing. The major RNA forms in the nucleolar fraction are 47S, 28S, and processed rRNA intermediates, whereas 18S rRNA is less abundant (Politz et al., *Mol. Biol. Cell*, 2003; Bai et al., *Proteomics*, 2012). Consistent with the previous results, 18S rRNA was barely detectable in the nucleolar fractions regardless of LPS stimulation, indicating that we successfully separated a nucleoplasmic fraction of high quality from the nucleoli (Please see the **additional Fig. #3**). Therefore, we are now confident that each fraction was successfully separated.

[redacted]

Data presented later in the manuscript (Figure 6b) further highlights the need to clarify the purity of these fractions. By immunofluorescence (Figure 5D), NCL is highly enriched in the nucleolus, consistent with most published data. However, western blotting of fractionated cells (figure 6B) shows equal distribution of NCL between nucleolar and nucleoplasmic fractions. How do the authors account for these discrepancies?

The reviewer raised a valid point by asking for clarity on the discrepancy of NCL localization between the immunoblots and the confocal data. According to many published reports on various cell lines, NCL is found predominantly in nucleoli, as in our IF results. Despite this fact, NCL is observed with immunoblot analysis in both nucleoplasm and nucleoli, and even in the cytosol, because NCL dynamically shuttles between the cytosol and the entire nucleus (Srivastava et al., *FASEB J.*, 1999; Ginisty et al., *J. Cell Sci.*, 1999; Mongelard et al., *Trends Cell Biol.*, 2007; Abdelmohsen et al., *RNA Biol.*, 2012). In accordance with many reports, we repeatedly observed that NCL was distributed sparsely throughout the nucleoplasm (Please see the enlarged images of Fig. 6d at the right of this response). This is presumably because the nucleolus represents a multiphase liquid condensate (Lafontaine et al., *Nat. Rev. Mol. Cell Biol.*, 2020), so the diffuse nucleoplasmic fluorescence signals are relatively weak while the nucleoli are intensely fluorescent. To avoid any confusion, we carefully reworded the text and added detailed information regarding the nucleolar fractionation in the Methods section of the revised manuscript.

• The following sentence is unintelligible: “Intriguingly, approximately 48% of detectable reads in the nucleolus were found to be protein-coding RNAs which exclusively possess introns, whereas fully spliced mRNAs were strictly limited to nucleoplasm to nucleolus transport and instead were rapidly translocated to the cytoplasm for translation (Fig 2a).” It seems like a clause has been deleted at some point. Could the authors please clarify what “exclusively possess introns” means and to what the percentages refer? Do the authors suggest that half of all pre-mRNA is found in the nucleolus? If so, this highlights the need for better analysis of fractionation. There is little evidence of such pervasive transcription and splicing of mRNAs in the nucleolus, which is the site of rRNA processing and ribosome assembly. Alternatively, do the authors mean to suggest that half of the “protein-coding” RNA in the nucleolus contains introns? This is what would be suggested by the above sentence, but is not true of what is represented in the data.

We thank the reviewer for catching this inappropriate statement interpreting the result. We carefully reworded the statement to properly represent the results shown in Fig. 2a.

We also realized that it was unclear whether the intronic RNAs enriched in the nucleoli represent intact pre-mRNAs or just intron remnants after splicing. To determine if the intronic reads were intact pre-mRNAs or just splicing byproducts with only introns, we selected major inflammatory and non-inflammatory genes and reanalyzed the RNA-seq dataset using the Integrative Genomics Viewer (IGV), which allowed us to determine the read densities of non-inflammatory and inflammatory genes in each fraction. Indeed, intact *Il6*, *Il1a*, and *Ccl2* pre-mRNAs, which exhibited both exons and introns, were enriched in the nucleolus (New Fig. 2a and New Supplementary Fig. 2c).

To avoid any confusion and to provide more convincing data, we also applied quantile normalization and reanalyzed the RNA-seq datasets, which allowed us to accurately assess global changes in distributions across different samples of each fraction. Consistent with the previous results, inflammatory pre-mRNAs were more enriched in the nucleolus compared with the nucleoplasm; however, this difference was not observed for non-inflammatory pre-mRNAs (New Figs. 2b and 2c). We have replaced the previous RNA-seq dataset with the new datasets analyzed with quantile normalization and incorporated the IGV results into the new Figs. 2a-c and new Supplementary Fig. 2c. We also revised the manuscript to reflect the reviewer's comments.

• With regards to figure 2a, it is also unclear what RNA is being analyzed. The figure legend states that it is a “comparison of changes in average exonic and intronic RNA levels of protein-coding genes...in CA, NP, and NL fractions at all time points of LPS.” So, is this only 2, 6, 12, and 15 hrs post LPS? Or is 0 hr included? If not, is the potentially perturbed distribution of exonic/intronic RNA a result of LPS treatment?

We agree with the reviewer's point about ambiguity in the legend for Fig. 2a. To avoid confusion, we carefully amended the legend and added more detailed information in the revised manuscript.

• RNA-FISH Data suggests that these RNAs are found at the periphery of the nucleolus, although, the area which was quantified obscures that fact. Are these mRNAs always found on the outer edge of the nucleolus?

As shown in the larger images below, the inflammatory *Il1b*, *Il6*, *Ccl2*, and *Cxcl2* pre-mRNA foci were preferentially positioned close to nucleoli or overlapped with the rim of the nucleolus, whereas non-inflammatory *Hmgal* pre-mRNA foci were positioned away from the nucleolus (New Figs. 3a-b; Please also see the Fig. #1 at the end of this response.). As mentioned above, we also tried to determine the spatial characteristics of the pre-mRNA foci, because several previous reports showed that actively transcribed regions are positioned away from the nucleolus, suggesting that sites located closer to the nucleolus tend to be transcriptionally inactive (Quinodoz et al., *Cell*, 2018; Padeken et al., *Curr. Opin. Cell Biol.*, 2014; Morf et al., *Nat. Biotech.*, 2019). We found that *Hmgal* pre-mRNA foci (a non-

inflammatory gene) were preferentially positioned in active RNA-pol II regions and excluded from the nucleolar compartment, whereas *Il6* and *Il1b* pre-mRNA foci (inflammatory genes) were positioned away from active RNA-pol II regions and overlapped with nucleoli. Although contacts between inflammatory foci and pS2 were slightly detectable in the nucleoplasm, we did not observe any in nucleolar regions. Importantly, the preferential organization of inflammatory pre-mRNA foci, but not that of non-inflammatory pre-mRNA foci, occurred in a LPS-dependent manner (**New Fig. 3d**; Please also see the **Fig. #2** at the end of this responses). We incorporated the updated results in the **new Figs. 3a, 3b, and 3d** in the revised manuscript.

The authors should comment on how this pertains to their previously published data regarding “InSac”. If maturation of IL-6 occurs in an “InSac” would that suggest that the InSac is a substructure within the nucleolus? In a previous publication, the InSac did not co-localize with the nucleolus, although looking at the data in that paper from the Park lab, they do appear to be peripheral to the nucleolus in the published figures. Is the InSac co-purifying with the nucleolus?

The reviewer raised an interesting question by asking if the InSAC is a substructure within the nucleolus. As shown in our previous results, InSAC was mostly located close to, but not overlapping, the nucleolus, unlike the inflammatory pre-mRNA foci that contacted or overlapped the nucleolus (**New Supplementary Fig. 3a**). Consistent with the confocal results, purified nucleoli did not contain TDP-43 (**New Supplementary Fig. 3b**). These results indicate that InSAC regions are positioned separately from the nucleolar region. Together with the results mentioned above, we believe that the nucleolus is a site for

inflammatory pre-mRNA decay rather than transcription or splicing. This is a good insight for further studies of the relationship between transcription, nucleoli, and inflammatory RNA foci, or between inflammatory RNA foci and InSAC during infection. Therefore, we have incorporated **new Supplementary Figs. 3a and 3b** and included additional statements in the revised manuscript to reflect the reviewer's points.

• **Similarly, the FISH data does not seem to correspond to the RNA-seq data. Granted, the meaning of the RNA-seq data is unclear, but if the interpretation is that half of the pre-mRNA is in the nucleolus and half is in the nucleoplasm (Fig 2A), this is not represented in the FISH data. How do the authors explain this discrepancy?**

The reviewer raised a valid point by asking for clarity on the discrepancy of inflammatory pre-mRNA localization between the FISH data and RNA-seq data. Although most mRNAs are known to be distributed in both the nucleoplasm and the cytosol, FISH signals for mRNAs are preferentially observed as speckles. This presumably is due to methodological specificity in FISH of the RNA probe conjugated with fluorescence. The fluorescent signals of nuclear condensates, such as nucleoli and speckles, therefore appear with relatively greater intensity than those of diffusely distributed in the entire nucleoplasm. In favor of this, NCL is known to be distributed in both nucleoplasm and nucleoli, and even in the cytosol, because it dynamically shuttles between the cytosol and the entire nucleus. Despite this fact, NCL is predominantly seen in nucleoli in IF analyses. In addition, many publications show that most of FISH signals representing mRNAs appear as a speckle-like pattern (Please see the additional Figures attached in the end of this response that are derived from references following; Shalek et al., *Nature*, 2014; Kwon et al., *Sci Rep.*, 2017; Samacoits et al., *Nat Commun.*, 2018; Yoon et al., *Nat Commun.*, 2013; Ben-Ari et al., *J Cell Sci.*, 2010).

[redacted]

***Note to Figures:** RNA-FISH showing the speckles of various inflammatory mRNAs (e.g. *Ifnb1*, *Ifit1*, *Cxcl1*, *Il6*, and *Tnf*) in LPS-stimulated bone-marrow-derived dendritic cells (BMDCs). The white arrow indicates the presence of both *Ifnb1* (magenta) and *Ifit1* (cyan) mRNAs in the BMDC. The gray lines delineate the border of cells.

• The authors suggest that it is possible that NCL is not necessary for rRNA processing in immune cells. This strains credulity without better data showing that is true. Ribosome biogenesis is an extremely conserved process not just amongst cell types, but between species. That every other cell needs NCL for rRNA maturation, but not immune cells is very tenuous. Particularly, since much of the early analysis of rRNA processing and transcription in mammalian cells were conducted using L1210 cells, which, while cancerous, were derived from immune cells (See the work of Dr. Sollner-Webb). Perhaps the discrepancy in the data arises from the fact that the KD of NCL is not as robust when analyzing rRNA (compare 2e with 2d). Given the wealth of information connecting NCL to rRNA maturation, the default position should be that NCL is required in immune cells, but the authors have not yet depleted it to a point where they affect rRNA maturation. If the authors believe that NCL is not essential for this function in immune cells, more solid data is required.

We agree with the reviewer's point that the current datasets are insufficient to clarify whether NCL is dispensable for rRNA processing in immune cells. As mentioned above, we provided additional results showing that neither 28S nor 47S rRNA levels were clearly altered in *Ncl*-knockdown cells (**New Supplementary Fig. 4e**). Nevertheless, we agree with the reviewer's concerns and have revised the text to reflect the reviewer's point.

• On a related note, other have shown that following LPS treatment, the transcription (and thus processing) of pre-rRNA increases >15x in B cells (Liu & Rose 1985, JBC). Thus, possibly in line with the authors supposition that NCL is not needed in immune cells, it is possible that resting cells do not require NCL, because they are not making much rRNA. This could explain the phenomena the authors see. However, this would present further complications in the interpretation of the data. Upon stimulation by LPS, ribosome biogenesis rapidly ramps up, at which point NCL becomes vital. Localization of RRP6 presented later (Figure 5) would also suggest this. Therefore, the loss of viability after LPS treatment in mouse experiments could be equally attributable to a defect in ribosome biogenesis after stimulation. Are the authors aware of any changes to the rate of rRNA synthesis (not the steady state levels) following LPS treatment in macrophages?

The reviewer raised a valid point by asking if alteration of rRNA expression occurs during LPS stimulation. To clarify the effect of LPS on rRNA expression levels, we performed an RT-qPCR assay with 47S rRNA-specific primers in macrophages throughout a time course of LPS stimulation (0, 2, 6, 12, and 15 h). We found no difference in the expression levels of 47S, 28S, and 18S rRNAs in response to LPS stimulation (**New Supplementary Fig. 2a**). Nevertheless, this gave us insight for further studies related to NCL function in rRNA processing in immune cells under different infection conditions.

• Based on location and sequence composition, the C/U rich-region that the authors

identify as being bound by NCL looks suspiciously like the “polypyrimidine tract” that is necessary for pre-mRNA splicing. Is that true or is this distinct from the polypyrimidine tract? If it is the polypyrimidine tract, wouldn’t that suggest that NCL may be involved in the splicing of these RNAs, either positively or negatively? That is, if NCL is bound to the polypyrimidine tract, it could compete with U2AF for binding and prevent its maturation thereby altering its metabolism.

We agree with the reviewer's comments that the C/U-rich sequences look similar to the polypyrimidine tract. The polypyrimidine tract is especially rich with uracil and is typically shorter than 20 nucleotides in length, located 5~40 base pairs before the 3' end of the intron to be spliced. However, the C/U-rich motif contains almost even numbers of cytidines and uracils and is usually over 20 nucleotides in length, located in various intronic positions. Thus, the C/U-rich motif is presumably different from the polypyrimidine tract. Indeed, as previously mentioned, we did not observe defects of inflammatory RNA splicing in *Ncl*-knockdown cells throughout many experiments. Nevertheless, this gave us insight for further studies related to the additional roles of NCL related to cooperation or competition with U2AF in inflammatory RNA stability or splicing. We have included additional statements in the Discussion section of the revised manuscript to reflect the reviewer's points.

• *The primer set used by the authors to analyze the 47S rRNA is not specific for the 47S rRNA. The primer pair amplifies a product that is entirely 5' of the A'/01 processing site. To truly analyze the 47S rRNA, this primer set should span this processing site. Otherwise, the authors could be analyzing cleaved 1-01 fragment (i.e. the 5' cleavage product following conversion of 47S to 45S). This is most relevant later regarding RRP6, which has been shown to be necessary for degradation of 5'ETS cleavage products (Kobylecki et al 2018). Since the authors functionally link NCL with RRP6 and argue that NCL aids in RRP6-mediated degradation this is particularly relevant.*

We agree with the reviewer's comment. We repeated the RT-qPCR using 47S rRNA-specific primers as the reviewer suggested. We provided this new data in the **new Supplementary Figs. 2a and 4e** of the revised manuscript.

• *The Actinomycin D experiment presented in figure 3E has a major confounding variable, if the authors are correct in their assumption that the nucleolar structure plays a role in the stability of these RNAs. ActD treatment rapidly causes the disruption of the nucleolus. In fact, pre-rRNA transcription is inhibited prior to pre-mRNA transcription due to the G-C content of rDNA. Loss of pre-rRNA transcription leads to the dissolution of the nucleolus. This also raises another confounding variable in that others have shown that siRNA knockdown of NCL alters nucleolar structure. If the authors premise, that the nucleolus is important for the stability of these RNAs, how do they distinguish between the specific effects of NCL and the perturbation of the nucleolus. Isn't this simply the same experiment as was conducted following NPM knockdown or etoposide treatment? How can the*

authors distinguish between the effect of NCL knockdown or transcriptional arrest when their argument regarding nucleolar pre-mRNA regulation relies on a stable nucleolus?

We agree with the reviewer's concerns that Act.D treatment or *Ncl* knockdown could possibly cause nucleolar disruption. To clarify this issue, we reexamined alterations in the size, morphology, and spatial preference of the nucleolus in *Ncl*-depleted cells. Consistently, we found no differences in the nucleolar structure, based on observations of fibrillarlin, after depletion of *Ncl*. Moreover, because 30~50% of total cells still expressed *Ncl* after the generation of stable *Ncl* shRNA expression, as mentioned in the previous response (Please see the red box in the highlighted regions of the **additional Figs. #1 and #2** at the end of this response), this residual expression might be enough to sustain NCL function in splicing and transcription. Indeed, we consistently observed no considerable detrimental effects of *Ncl* knockdown on cell viability, survival, splicing, transcription, or apoptosis throughout all our experiments.

• **The localization of Rrp6 is interesting and certainly suggests a dynamic role for this protein following LPS treatment. Previous work has shown that this protein is also necessary for rRNA processing. In fact, the name for the protein is “ribosomal RNA processing.” Thus, it is confusing as to why Rrp6 is not localized to the nucleolus under basal conditions, as it is in other cell types. Other studies have demonstrated that this protein is localized to the nucleolus and has multiple roles in the maturation of rRNAs. In addition to NCL not being necessary for rRNA processing in immune cells, do the authors suggest that Rrp6 is also dispensable? Or is it not necessary for rRNA maturation until after the cells have been stimulated?**

The reviewer raised a valid point by asking why Rrp6 is not observed in the nucleolus despite its functions in rRNA processing. As shown in Fig. 6b, Rrp6 was detectable in both the nucleoplasm and the nucleolus in immunoblot analyses of cellular fractions. We should therefore ask for the reason for the discrepancy in NCL localization between the immunoblots and the confocal data. As mentioned above, NCL was found predominantly in nucleoli in our IF analysis, although it is known to be observed in both nucleoplasm and nucleoli, and even in the cytosol, in immunoblot analyses, because it dynamically shuttles between the cytosol and the entire nucleus (Srivastava et al., *FASEB J.*, 1999; Ginisty et al., *J. Cell Sci.*, 1999; Mongelard et al., *Trends Cell Biol.*, 2007; Abdelmohsen et al., *RNA Biol.*, 2012). This is

presumably because the nucleolus represents a multiphase liquid condensate (Lafontaine et al., *Nat. Rev. Mol. Cell Biol.*, 2020) and therefore exhibits fluorescent signals of relatively high intensity. In fact, enlarged images show Rrp6 in the nucleolus, albeit with dim fluorescent signals. (Please see the **enlarged images of new Fig. 6f** at the end of this response.) To avoid any confusion, we carefully reworded the text of the revised manuscript.

- **Could the authors clarify what is being measured in the 5th set of columns in figure 5c? Each set of columns are Il1b and Il6 pre-mRNA and mRNA. The 5th is identified as “k/d”. Is this a knockdown of each individual RNA?**

We thank the reviewer for identifying this ambiguity. We amended the labels in the revised manuscript.

- **The data concerning the dependence of NCL:RRP6 interaction on sumoylation (figure 6F) is weak. Following LPS treatment, there is still pulldown of RRP6 with K296R-NCL, although, I agree, it does seem weaker. However, this must be quantified over multiple experiments to be convincing.**

We agree with the reviewer's point. We provided additional graphs obtained by quantifying the NCL–Rrp6 binding affinity in the **new Fig. 7e** of the revised manuscript.

REVIEWERS' COMMENTS

Reviewer #1 (Remarks to the Author):

Concerns raised by this referee have been satisfactorily addressed.

Reviewer #2 (Remarks to the Author):

The authors carefully addressed my concerns and significantly strengthened the evidence for their model. With this study, the authors make the exciting and intriguing discovery that nucleoli are sites where inflammatory pre-mRNAs are specifically degraded during macrophage activation, thereby assigning a novel function to nucleoli.

Reviewer #3 (Remarks to the Author):

The authors have done a remarkable job responding to most of my criticisms. Their revisions have brought much clarity to the text and have strengthened their arguments. The additional experiments are done to the same high standard as the original experiments. I commend them for highlighting some weaknesses in the text, a topic that far too many researchers are reluctant to do. I may have some slight "quibbles" with the interpretations, but these are better suited for a discussion, rather than in additional revisions. Convincing.

Because we have successfully addressed all remaining concerns and there have been no further comments, we only uploaded the revised manuscript to address the editorial requests.

REVIEWERS' COMMENTS

Reviewer #1 (Remarks to the Author):

Concerns raised by this referee have been satisfactorily addressed.

Reviewer #2 (Remarks to the Author):

The authors carefully addressed my concerns and significantly strengthened the evidence for their model. With this study, the authors make the exciting and intriguing discovery that nucleoli are sites where inflammatory pre-mRNAs are specifically degraded during macrophage activation, thereby assigning a novel function to nucleoli.

Reviewer #3 (Remarks to the Author):

The authors have done a remarkable job responding to most of my criticisms. Their revisions have brought much clarity to the text and have strengthened their arguments. The additional experiments are done to the same high standard as the original experiments. I commend them for highlighting some weaknesses in the text, a topic that far too many researchers are reluctant to do. I may have some slight "quibbles" with the interpretations, but these are better suited for a discussion, rather than in additional revisions.